# A CYBDOM protein impacts iron homeostasis and primary root growth under phosphate deficiency in Arabidopsis

Joaquín Clúa [1], Jonatan Montpetit[1], Pedro Jimenez-Sandoval [1], Christin Naumann [2], Julia Santiago [1] & Yves Poirier [1]✉

Arabidopsis primary root growth response to phosphate (Pi) deficiency is mainly controlled by changes in apoplastic iron (Fe). Upon Pi deficiency, apoplastic Fe deposition in the root apical meristem activates pathways leading to the arrest of meristem maintenance and inhibition of cell elongation. Here, we report that a member of the uncharacterized cytochrome b561 and DOMON domain (CYBDOM) protein family, named CRR, promotes iron reduction in an ascorbate-dependent manner and controls apoplastic iron deposition. Under low Pi, the *crr* mutant shows an enhanced reduction of primary root growth associated with increased apoplastic Fe in the root meristem and a reduction in meristematic cell division. Conversely, *CRR* overexpression abolishes apoplastic Fe deposition rendering primary root growth insensitive to low Pi. The *crr* single mutant and *crr hyp1* double mutant, harboring a null allele in another member of the CYDOM family, shows increased tolerance to high-Fe stress upon germination and seedling growth. Conversely, *CRR* overexpression is associated with increased uptake and translocation of Fe to the shoot and results in plants highly sensitive to Fe excess. Our results identify a ferric reductase implicated in Fe homeostasis and developmental responses to abiotic stress, and reveal a biological role for CYBDOM proteins in plants.

Phosphorus (P) and iron (Fe) are both essential elements for plants. P is found in cells in the form of orthophosphate ($PO_4^{-3}$) and is included in nucleic acids, proteins, lipids, and numerous other small molecules playing essential roles in energy metabolism, such as adenosine triphosphate (ATP)[1]. Fe cycles between two oxidation states, $Fe^{3+}$ (ferric) and $Fe^{2+}$ (ferrous), and participates in numerous fundamental processes and pathways involving redox reactions, including respiration and photosynthesis[2]. Although typically abundant in most soils, both P and Fe are largely present in insoluble forms, with P often forming complexes with Fe (hydro)oxides. In most soils, the amount of soluble inorganic phosphate (Pi; $H_2PO_4^-$ or $HPO4^{-2}$), the form of P acquired by plant roots, is typically 0.5–10 μM, while the concentration of soluble

Fe in well-aerated soils can be as low as 100 pM in calcareous soils[3,4]. In waterlogged soils with an acidic pH and low redox potential, as are often encountered in many wetland rice (*Oryza sativa*)-growing areas, high levels of $Fe^{2+}$ can lead to iron toxicity[5]. In all plants, except members of the Gramineae, the main strategy to acquire Fe through the roots involves three main steps: (1) $Fe^{3+}$ solubilization from the soil via acidification, implicating $H^+$-ATPase pumps such as AHA2, as well as the release of low molecular weight organic compounds capable of chelating $Fe^{3+}$, such as malate, flavins, and coumarins; (2) the reduction of soluble $Fe^{3+}$ to $Fe^{2+}$ by the plasma membrane ferric reductase FRO2; and (3) the import of $Fe^{2+}$ via the IRT1 transporter[6–9]. Similarly, the acquisition of Pi from P–Fe insoluble complexes also involves Pi

¹Department of Plant Molecular Biology, Biophore Building, University of Lausanne, 1015 Lausanne, Switzerland. ²Department of Molecular Signal Processing, Leibniz Institute of Plant Biochemistry, Weinberg 3, 06120 Halle, Germany. ✉e-mail: yves.poirier@unil.ch

solubilization via medium acidification and the release of organic acids (citrate, malate), followed by H⁺–Pi co-transport by PHT1 transporters[10].

Numerous interactions have been described in the pathways involving Fe and Pi homeostasis[11]; for example, Pi deficiency was shown to result in an increased plant Fe content[12–14] and the downregulation of numerous genes involved in Fe acquisition, such as *IRT1* and *FRO2*, while at the same time increasing the expression of other genes, such as *FER1* and *YSL8*, implicated in Fe storage and transport, respectively[13,15–17]. Conversely, Fe deficiency can result in Pi accumulation[18,19]. Strong evidence indicates that this pattern of co-regulation is not simply the result of the increased availability of Fe under Pi-deficient conditions, and vice-versa, comes from the implication of two central regulators of Pi and Fe homeostasis. PHR1 and its close homolog PHL1 are members of the GARP family of transcription factors and are responsible for the majority of the transcriptional changes occurring under Pi deficiency[20]. Several of the Fe-homeostasis genes activated by Pi deficiency, such as *FER1*, are direct targets of PHR1[21,22]. In the Fe pathway, members of the Brutus family of E3 ligases (BTS, BTSL1, and BTSL2 in *Arabidopsis thaliana* and HRZ, HRZ2, and HRZ3 in rice) participate in the degradation of key transcription factors involved in Fe homeostasis, such as FIT in *A. thaliana* and PRI in rice, and are also thought to act as Fe sensors[23–27]. A recent study in rice has shown that HRZ1 and HRZ2 interact with and mediate the degradation of PHR2 (the rice orthologue of PHR1), explaining why Fe deficiency can attenuate the Pi deficiency response[28]. Furthermore, the same study found that PHR2 transcriptionally downregulates the expression of the HRZs. PHR1/2 and BTS/HRZ thus form a reciprocal inhibitory module contributing to the coordination of Pi and Fe homeostasis.

Interactions between Pi and Fe also influence root growth[29]. While primary root growth is inhibited under either Pi or Fe deficiency, combined Fe and Pi deficiency results in a reversion of this growth suppression. Genetic screens involving mutants in the response of primary root growth under Pi deficiency have identified several genes involved in the Fe–Pi interaction during root growth. *LPR1* encodes a ferroxidase that converts $Fe^{2+}$ to $Fe^{3+}$, while *PDR2* encodes an endoplasmic reticulum (ER)-localized P5-type ATPase thought to negatively affect LPR activity via an unknown mechanism[30–33]. While the primary root growth of the *lpr1* mutant is unaffected by Pi deficiency, the *pdr2* mutant is hypersensitive under the same condition[31,32,34]. Additional components of the pathway include the malate efflux channel ALMT1; the STOP1 transcription factor, which regulates *ALMT1* expression; the ALS3 and STAR1 subunits, which together form a tonoplast ABC transporter complex; and the CLE14 peptide receptors CLV2 and PEPR2[35–38]. The current model proposes that, under Pi-deficient conditions, malate secretion mediated by ALMT1 chelates $Fe^{3+}$, allowing its accumulation in the apoplast of the root meristem and elongation zones, where it inhibits cell elongation and division. Apoplastic $Fe^{3+}$ would participate in a redox cycle implicating the LPR1 ferroxidase, generating reactive oxygen species (ROS) and leading to a reduction in cell-to-cell communication via callose deposition, meristem differentiation by the activation of the CLE14–CLV2/PERP pathway, and a reduction in cell wall extensibility by the action of peroxidases[35,39,40]. Blue light has also been implicated in this pathway, both as a cryptochrome-mediated signal transmitted to the roots and as a catalyst in Fenton reactions[41,42]. Potential role for ferric reductase participating in this redox cycle is yet to be elucidated[39,43,44].

We recently identified the ER chaperones Calnexin1 (CNX1) and Calnexin2 (CNX2) as components of the Fe-dependent inhibition of primary root growth under Pi deficiency, as the *cnx1 cnx2* double mutant shows greater Fe-dependent reduction of primary root growth under Pi deficiency compared to Col-0[45]. A proteomic analysis of the *cnx1 cnx2* double mutant identified a reduction in the amount of an uncharacterized CYB561 protein with a N-terminal DOMON domain,

hereafter named CRR. Here, we show that CRR encodes a plasma membrane-localized ferric reductase that participates in Fe homeostasis as well as in the control of primary root growth under Pi deficiency.

## Results

### CRR encodes a CYBDOM protein that is less abundant in the *cnx1 cnx2* double mutant

In a previous work, we showed that the *A. thaliana cnx1 cnx2* double mutant showed a reduction in primary root growth under Pi deficiency, which was associated with an increased apoplastic Fe accumulation at the root meristem[45]. Since the CNX proteins are components of the ER protein folding and quality control machinery, we hypothesized that the root phenotype of the *cnx1 cnx2* mutant could be a consequence of the misfolding, degradation, and/or destabilization of a particular set of proteins. To test this hypothesis, we performed a proteomic analysis of Col-0 and *cnx1 cnx2* mutant roots grown for 7 days in Pi-sufficient or -deficient conditions (+Pi or −Pi, respectively). In the +Pi condition, there were only 18 differentially abundant proteins between Col-0 and the *cnx1 cnx2* mutant, whereas in the −Pi condition, there were 136 differentially abundant proteins (Supplementary Data 1 and Supplementary Fig. 1a). The higher number of differentially abundant proteins in the −Pi treatment likely reflects the fact that the *cnx1 cnx2* mutant only showed a root growth phenotype under Pi deficiency[45]. A Gene Ontology (GO) analysis of the differentially abundant proteins revealed they were enriched in functions related to ER stress, protein repair, and the response to unfolded proteins, indicating a predominant effect on ER protein homeostasis in the *cnx1 cnx2* mutant (Supplementary Fig. 1b).

To identify candidate genes implicated in the primary root growth phenotype of the *cnx1 cnx2* mutant, we focused on proteins that were differentially abundant in both the +Pi and −Pi conditions. Among the seven proteins differentially expressed in both conditions, there was a member of the CYBDOM family (AT3G25290) that we named CRR for CYBDOM ROOT REDUCTION (Supplementary Fig. 1c). As a typical CYBDOM protein, CRR contains a signal peptide at its N-terminal end, followed by a DOMON domain (IPR005018) and a cytochrome b561 (CYB561) (PS50939) (Fig. 1a and Supplementary Fig. 2a). CYBDOM proteins have unknown functions in plants, and the *A. thaliana* genome encodes 10 of them[46]. Notably, the mouse and human genomes each contain a single CYBDOM protein, named SDR2, and expression of the mouse SDR2 in *Xenopus* oocytes revealed $Fe^{3+}$ reductase activity[47]. As expected for CYBDOM members, the CYB561 domain of CRR contains five transmembrane alpha helices as well as four conserved histidine groups involved in coordinating two heme *b* molecules[48] (Fig. 1a, Supplementary Fig. 2a, b). The N-terminal DOMON domain is predicted to be located in the apoplast and contains a conserved methionine and histidine previously shown to be involved in coordinating another *b* heme group in the cellobiose dehydrogenase of the fungus *Phanerochaete chrysosporium*[48,49] (Fig. 1a, Supplementary Fig. 2a, b).

In order to obtain a better understanding if all the functional elements of DOMON and CYB561 domains are conserved in CRR, we used the predicted structure from the AlphaFold database (Supplementary Figs. 3 and 4) to perform a structural comparative analysis with other DOMON and CYB561 containing proteins. The model shows that Met89 and His180 of the DOMON domain, as well as histidine pairs 220–289 and 256–325 of the CYB561 domain are conserved and properly oriented to coordinate three potential *b* heme moieties (Supplementary Figs. 3a and 4). Moreover, the superimposition of the CRR predicted model with the crystal structures of the *P. chrysosporium* cellobiose dehydrogenase DOMON domain (PDB ID:1D7B) and Arabidopsis CYB-1 cytochrome b561 domain (PDB ID: 4O79), shows a high degree of conservation in terms of overall structure and electrostatic potential surface (Supplementary Fig. 3b–d). Analysis of CYB-

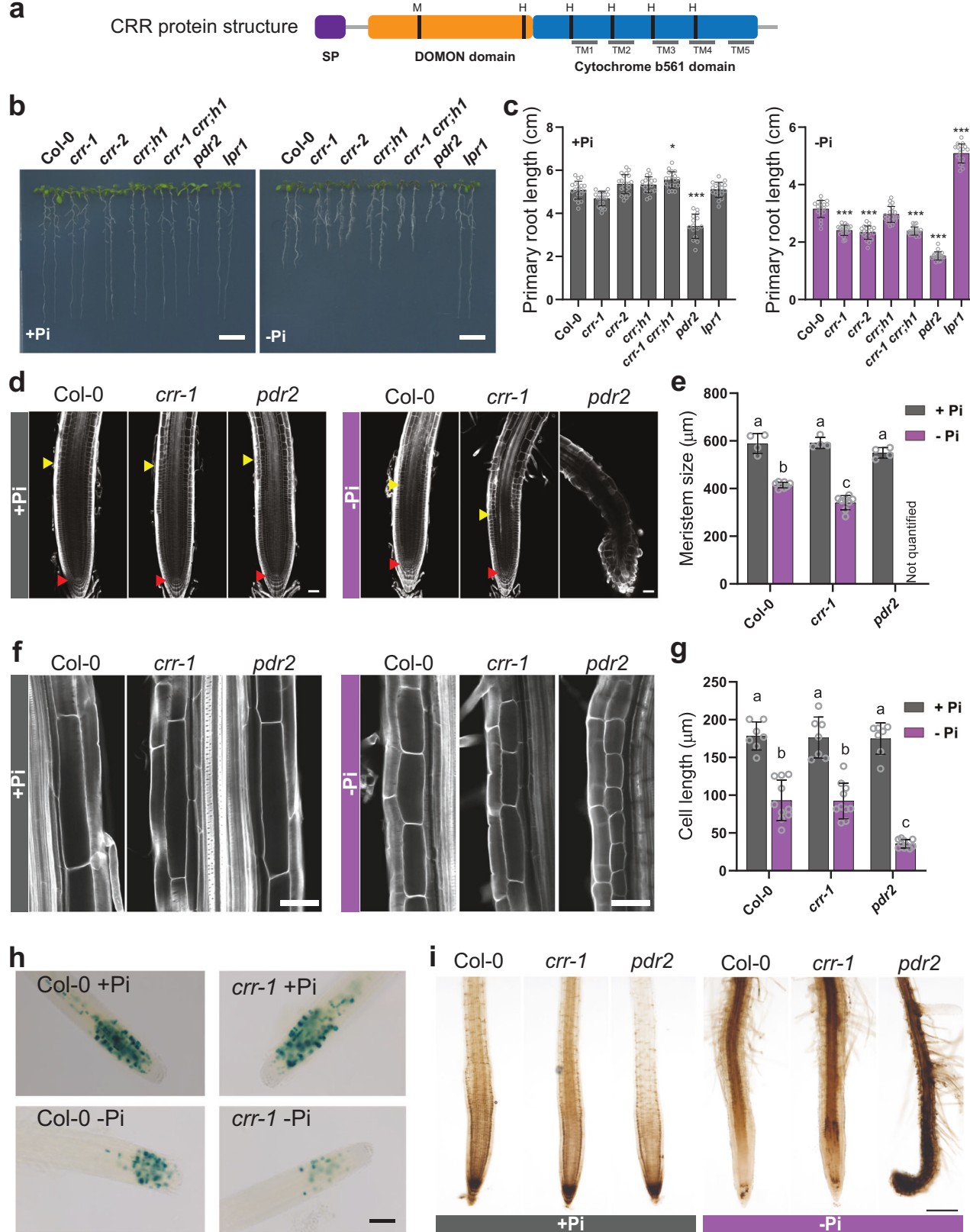

1 crystal structure revealed that ascorbate binds to a positively charged pocket in the cytosolic region of the protein. Structural analysis of the CYB561 domain of the CRR protein unveiled the presence of an analogous positive patch, containing key residues for ascorbic acid coordination (Supplementary Fig. 3b–f). This structural study strongly suggests that the CRR CYB561 domain could bind ascorbate in the

cytosol, and potentially transfer an electron through the three *b* heme moieties to reduce iron in the apoplast.

CRR has previously been detected in proteomic experiments analyzing purified plasma membranes, as reported on the SUBA database (https://suba.live/). To validate these predictions, we first analyzed the subcellular localization of a CRR-GFP translational fusion

**Fig. 1 | CRR encodes a CYBDOM protein involved in primary root growth under phosphate deficiency. a** Diagram of the CRR protein structure showing the positions of the signal peptide (SP), the DOMON and cytochrome b561 domains, the five transmembrane (TM) alpha helices, and the conserved methionine (M) and histidine (H) residues. **b** Phenotypes of Col-0, *crr-1*, *crr-2*, *crr;h1*, *crr-1 crr;h1*, *pdr2*, and *lpr1* seedlings grown for seven days in Pi-sufficient (+Pi) or Pi-deficient (−Pi) conditions. Scale = 1 cm. **c** Quantification of the primary root length at seven days after germination in +Pi and −Pi. The statistical differences were analyzed using a one-way ANOVA followed by Tukey's multiple comparisons test. Significant differences to Col-0 are indicated (***$P < 0.001$; **$P < 0.01$; *$P < 0.05$). **d** Root apical meristem (RAM) structure of Col-0, *crr-1*, and *pdr2*. Red arrows indicate the quiescent center and yellow arrows indicate the end of the meristematic zone. **e** Quantification of the size of the RAM. For each genotype $n = 4$ and 8 for +Pi and −Pi, respectively. **f** Representative images showing the lengths of the cells in the root differentiation zone. In (**d**) and (**f**), the root cell walls were stained with Calcofluor White and visualized using confocal microscopy. Scale = 50 μm. **g** Quantification of the lengths of the differentiated root cells of plants grown in +Pi and −Pi. For each genotype $n = 7$ and 10 for +Pi and −Pi, respectively. In (**c**), (**e**) and (**g**), the data are presented as means ± sd and the circles represent the number of values analyzed. The statistical differences were analyzed using a one-way ANOVA in (**e**) and a two-way ANOVA in (**g**), followed by Tukey's multiple comparisons test. Different letters indicate significant differences ($P < 0.05$). **h** GUS staining of Col-0 and *crr-1* carrying the G2/M transition marker pCYCB1::GUS. Scale = 100 μm. **i** Visualization of the iron distribution using the Perls-DAB staining of plants grown in +Pi and −Pi. Scale = 200 μm. For (**h**) and (**i**), data from an independent experiment is shown in Supplementary Figs. 10 and 11, respectively. Source data for (**c**), (**e**) and (**g**), including *n* and *p* values, are provided as a Source data file.

protein expressed under the CaMV *35S* promoter in *A. thaliana*. The confocal microscopy images reveal that CRR is mainly localized at the plasma membrane with a small fraction at the ER (Supplementary Fig. 5a). To further validate this localization, we transiently expressed CRR-GFP in *Nicotiana benthamiana* leaves together with a plasma membrane (CBL1-OFP)[50] or ER (ER-rk-RFP)[51] marker. CRR-GFP mostly co-localized with the plasma membrane marker, including during plasmolysis, with a small fraction associated with the ER marker, suggesting that the protein transits the secretory pathway from the ER to the plasma membrane (Supplementary Fig. 5b).

The topology of CRR was assessed using a split-GFP assay[52]. The C terminus of CRR was fused to the 11th beta-strand of the GFP protein (CRR-GFP11) and co-expressed with a cytosolic or secreted version of the remaining 10 GFP beta-strands (GFP1–10). GFP fluorescence was significantly restored when CRR-GFP11 was co-expressed with the cytosolic GFP1–10, confirming that the CRR C terminus faces the cytosol (Supplementary Fig. 5c). Considering the five transmembrane domains of the CYB561 domain, the DOMON domain is thus apoplastic.

## CRR is a negative regulator of the primary root growth response to Pi deficiency

The *cnx1 cnx2* double mutant showed a stronger reduction of primary root growth in response to Pi deficiency than in Col-0[45]; therefore, we assessed the root phenotype of the *crr* mutants under the same condition. The *crr-1* (*Salk_202530*) and *crr-2* (*Salkseq_039879.2*) mutants contain a T-DNA insertion in the first intron and second exon, respectively. A qRT-PCR analysis confirmed that both mutants had strongly reduced *CRR* mRNA levels in comparison with the wild-type (WT) plants grown in +Pi or −Pi conditions (Supplementary Fig. 6a, b). Furthermore, full-length *CRR* cDNA was undetectable by PCR in these mutants, indicating that both of them are knockout mutants (Supplementary Fig. 6c).

The primary roots of *crr-1* and *crr-2* were compared with Col-0 and the previously characterized *lpr1* and *pdr2* mutants after 7 days of growth on −Pi and +Pi media[31,34] (Fig. 1b, c). As previously reported, the *pdr2* mutant showed pleiotropic growth defects in both +Pi and −Pi conditions, including a strong inhibition of primary root growth in the −Pi condition. On the other hand, the *lpr1* primary roots were completely insensitive to Pi deficiency, producing roots of equal length to those grown in the +Pi medium. The *crr-1* and *crr-2* mutant roots were approximately 25% shorter than those of Col-0 when grown in −Pi medium, but no difference was observed in the +Pi medium (Fig. 1b, c). Similar to the *cnx1 cnx2* double mutant and *pdr2*, the short-root phenotype of the *crr* mutants under −Pi was dependent on the presence of Fe in the medium, as the primary root growth of the *crr-1* mutant in a −Pi −Fe medium was similar to that of Col-0 (Supplementary Fig. 7).

The *A. thaliana* CYBDOM protein family is divided into four different clades[46]. CRR belongs to a clade of six members, and its closest homologous gene is AT4G12980, which was named CRR homolog 1

(*CRR;H1*) (Supplementary Fig. 2c–d). The high degree of identity between these two proteins could imply functional redundancy. To test this hypothesis, we compared the primary root growth of the *crr-1* and *crr;h1* single mutants, the *crr-1 crr;h1* double mutant, and Col-0 under both +Pi and −Pi conditions. No significant differences were observed between *crr;h1* and Col-0 or between the *crr-1 crr;h1* double mutant and the *crr-1* single mutant, leading to the conclusion that CRR;H1 is not involved in the primary root responses to Pi deficiency (Fig. 1b, c). In an accompanying manuscript Maniero et al.[53] reported that another CYBDOM member named HYP1 (AT5G35735) plays a role in root development under Pi deficiency. To assess the relative contribution of each gene to root elongation, we analyzed *crr*, *hyp1*, and *crr hyp1* phenotypes under +Pi and −Pi. Similar to *crr*, *hyp1* and the double mutant did not show any defect in primary root growth under +Pi condition. However, under −Pi, *hyp1* showed a stronger inhibition of root development than *crr*, and the *crr hyp1* was significantly more affected than the single mutants (Supplementary Fig. 8). Even though the double mutant phenotype under −Pi was milder than *pdr2*, it did not show the pleiotropic defects that *pdr2* shows in +Pi condition. Altogether, these results suggest that CRR and HYP specifically regulate root growth under Pi deficiency in a partially redundant manner.

Given that *crr-1* and *crr-2* are both T-DNA insertional mutants, we wanted to rule out the possibility that the observed root phenotype was a consequence of changes in the expression of *CRR*-neighboring genes. For this reason, we decided to edit the *CRR* locus using CRISPR/Cas9 technology. After generating transgenic lines expressing two different sets of guide RNAs, we obtained two independent homozygous lines in which *CRR* was edited. Line 25.6 has an in-frame deletion of 1120 nucleotides, generating an ORF potentially encoding a peptide of 93 aa, whereas line 22.6 has a deletion of 729 nucleotides, generating a hypothetical truncated version of CRR lacking the whole CYB561 domain (Supplementary Fig. S9a). When lines 25.6 and 22.6 were grown in +Pi and −Pi media, we observed a reduction in primary root growth relative to Col-0 under the −Pi condition only (Supplementary Fig. 9b). Together, these data show that *CRR* contributes to the primary root growth response under Pi deficiency.

## The short-root phenotype of *crr* is a consequence of defects in the root apical meristem maintenance associated with a hyperaccumulation of Fe

Primary root growth is essentially determined by cell division in the apical meristem and cell elongation in the root elongation zone. An inspection of the meristem morphology of Col-0 roots grown in −Pi showed the typical reduction in meristem size and the loss of identity of the quiescent center (QC) compared to roots grown in +Pi (Fig. 1d, e). The general organization and morphology of the meristem of *crr-1* was similar to the Col-0 roots when grown under −Pi, including a loss of QC identity; however, the reduction in meristem size was exacerbated, resulting in a 21% smaller meristem than that of Col-0 (Fig. 1e). In

contrast, the *pdr2* mutant showed a very strong disorganization of the entire meristematic region, with large swollen cells.

Inspection of the root differentiation zone showed a 50% reduction in cell length in the Col-0 roots grown on −Pi compared with those on +Pi (Fig. 1f, g), while in *pdr2*, this response was more pronounced, resulting in cell lengths reaching just 38% of those in Col-0 under −Pi. By contrast, *crr-1* showed no significant differences in cell length compared with those of Col-0 in the +Pi or −Pi conditions (Fig. 1f, g). The CYCB1::GUS cell cycle reporter cassette was introgressed into the *crr-1* background to visualize meristematic cell division activity (Fig. 1h and Supplementary Fig. 10). As previously reported, the meristematic cell division activity of Col-0 plants grown in −Pi was reduced when compared with the +Pi treatment[34]. However, cell division activity under the −Pi condition was even more reduced in *crr-1* compared with Col-0, while *crr-1* cell division activity was not affected under the +Pi condition. Taken together, these results reveal that CRR acts as a positive regulator of root apical meristem division and maintenance under Pi deficiency.

As the hypersensitive primary root growth inhibition of the *crr* mutant under the −Pi condition is dependent on Fe, we next explored the distribution of Fe in the root tips of *crr-1* and *pdr2* using Perls-DAB staining. In the +Pi condition, all the tested genotypes showed the strongest Fe accumulation at the root cap, consistent with previous reports[38,40] (Fig. 1i and Supplementary Fig. 11). The histochemical detection of iron using the Perls staining without the DAB intensification step (Perls staining), further confirmed this iron distribution pattern (Supplementary Fig. 12). Under Pi deficiency, Perls-DAB staining showed that Col-0 had a reduced level of Fe at the root cap and an accumulation at the stem cell niche (SCN, which includes QC and initials), as well as at the elongation and differentiation zones. Notably, *crr-1* roots followed the same Fe distribution pattern as Col-0, albeit with higher intensity (Fig. 1i and Supplementary Fig. 11). In agreement with its stronger root growth inhibition, *pdr2* showed a high accumulation of Fe in all root zones under Pi deficiency, which was also detectable without the DAB intensification step (Fig. 1i, Supplementary Figs. 11 and 12). Taken together, our results show that *crr* accumulates more Fe than Col-0 under Pi deficiency, reducing cell division in the meristem. Importantly, while Fe accumulation in the elongation zone of the *crr* mutant was higher than in Col-0, it did not affect cell elongation.

A qRT-PCR analysis revealed an induction of *CRR* expression in the shoots under −Pi, while expression in roots was unaffected by the Pi status (Supplementary Fig. 13a). For unknown reasons, all experiments aimed at analyzing the endogenous *CRR* expression pattern using either *CRR* promoter fragments ranging from 200 to 5000 bp fused to GUS or gene fragments including the *CRR* promoter and *CRR* genomic coding regions or cDNA fused to GFP failed to reveal GUS or GFP expression, respectively. These observations indicate that the regulatory context for *CRR* expression might be more complex than typically observed for most *A. thaliana* genes and that important features, such as higher-order chromatin organization or epigenetic signatures, may play an important role in *CRR* promoter activity that cannot be reproduced in a transgenic setting[54,55]. Single-cell transcriptomic data from *A. thaliana* roots revealed the expression of *CRR* in the SCN, as well as in differentiating cortical, endodermal, and lateral root cap cells[56] (Supplementary Fig. 13c, d). The co-expression of *CRR* with *pPUB25* and *pSPT* further indicated the expression of *CRR* in the root meristem (Supplementary Fig. 13b)[57].

To identify the cells of the root tip in which the expression of *CRR* is required to complement the *crr* mutant phenotype, a *CRR-GFP* translational fusion construct was expressed under the control of promoters active in various regions of the root, including the QC, endodermis, vasculature, cortical cells, and root cap (Supplementary Fig. 14a). Confocal microscopy images confirmed that the *LOVE1*, *SCR*, *SHR*, *WOX5*, *WOL*, *CO2*, *PEP*, *ATL75*, *MYB36*, and *LPR1* promoters

allowed expression of *CRR-GFP* in their corresponding root domains under +Pi conditions (Supplementary Fig. 14b, upper panel). Under −Pi condition, these promoters maintained the expected expression pattern, with the notable exception of WOX5, which lost its QC-specific expression pattern, likely as a result of QC cell differentiation observed under such condition[40] (Supplementary Fig. 14b, lower panel). Only transgenic plants expressing *CRR-GFP* under the control of the *LPR1* promoter (*pLPR1::CRR-GFP*), enabling strong expression in the SCN and weaker expression in cells layers of the meristematic zone including the endodermis and cortical cells, were able to complement the *crr-1* mutant (Supplementary Fig. 15). In these lines, CRR-GFP is located at the PM, further confirming the subcellular localization of CRR (Supplementary Fig. 14c). It is clear that narrow *CRR-GFP* expression only in the pericycle initials (pATL75), meristematic vasculature (pSHR, pWOL), meristematic endoderm (pSCR) or meristematic cortical cells (pCO2) failed to rescue the mutant phenotype. Furthermore, expression of *CRR-GFP* in the QC mediated by the WOX5 promoter under +Pi condition was not sufficient to prevent lost of QC identity and reduction in meristem cell division observed upon −Pi condition. Together, these results indicate that *CRR* expression in the *LPR1* domain, i.e. at the SCN and more broadly in the meristematic cell layers encompassing the endoderm and cortical cells, is likely key to ameliorating root growth inhibition under low Pi. Such conclusion would be supported by the pattern of Fe accumulation observed in the *crr* mutant under −Pi condition (Fig. 1i).

## Primary root growth is not inhibited by Pi deficiency in plants overexpressing *CRR*

To assess the effects of *CRR* overexpression on root growth, we analyzed transgenic lines expressing a *CRR-GFP* translational fusion construct (CRR OE) under the control of the CaMV *35S* promoter in both Col-0 and the *crr-1* mutant. Under +Pi conditions, the ectopic expression of *CRR-GFP* did not cause any obvious phenotype regarding shoot growth or root development in the Col-0 or *crr-1* backgrounds; however, under −Pi, *CRR* overexpression in both Col-0 and *crr-1* made primary root growth completely insensitive to Pi deficiency (Fig. 2a, b). In our experimental conditions, both the CRR OE line and *lpr1* showed a similar primary root growth that is unaffected by the Pi supply (Fig. 2a, b). Considering the expression pattern of the endogenous *CRR* in roots (Supplementary Fig. 13c, d), the phenotypes of CRR OE lines is likely due to either the higher level of expression attained in root cells normally expressing *CRR* and/or its ectopic nature.

Meristem size and the length of cells in the differentiation zone of CRR OE were not significantly different in roots grown on +Pi or −Pi media, explaining its resemblance to the *lpr1* mutant (Fig. 2c, d, f and Supplementary Fig. 16). A closer inspection of the QC showed that, in Col-0 roots, the identity of the QC was lost under Pi deficiency and cells underwent differentiation, while the QC of the CRR OE roots grown in −Pi remained defined and undifferentiated as in the +Pi condition (Fig. 2e). The same effect on the QC was observed for *lpr1*.

Analysis of Fe deposition by Perls-DAB staining showed that CRR OE had weaker Fe staining than Col-0 in the root cap under +Pi, but still more than *lpr1* (Fig. 2g and Supplementary Fig. 17). In the −Pi condition, CRR OE and *lpr1* showed minimal Fe staining in the SCN and meristematic zone. Altogether, these results suggest that CRR OE root growth is insensitive to Pi starvation as a consequence of a disruption in Fe accumulation at the root tip.

## *LPR1* and *PDR2* are epistatic to *CRR*

The genetic interactions between *CRR* and the main molecular players of the primary root responses to Pi deficiency were assessed by crossing *crr-1* with the long-root mutants *almt1* and *lpr1* and analyzing their root phenotypes under the +Pi and −Pi conditions. In +Pi, the root lengths of all lines were similar, with no obvious differences in terms of architecture (Fig. 3a, b). In agreement with the macroscopic

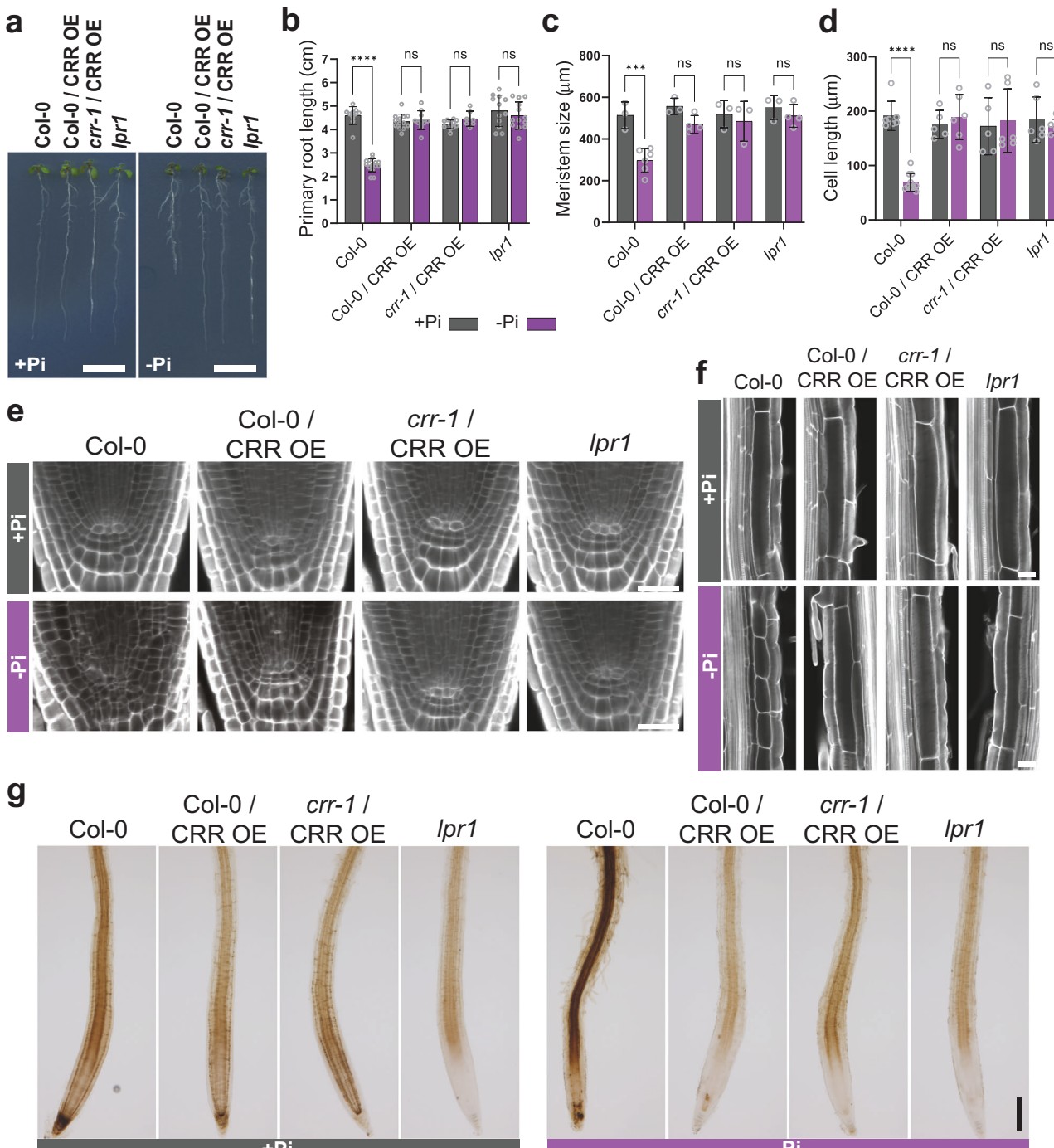

**Fig. 2 | *CRR*-overexpressing plants are insensitive to phosphate deficiency-induced reductions in primary root growth. a** Phenotypes of the transgenic lines expressing *CRR-GFP* under the control of the *35S* promoter in the Col-0 (Col-0/CRR OE) or *crr-1* (*crr-1*/CRR OE) backgrounds, as well as Col-0 and *lpr1*, grown on Pi-sufficient (+Pi) or Pi-deficient (−Pi) media for seven days. Scale = 1 cm. The roots of all lines were analyzed for **b** primary root length; **c** meristem size; and **d** cell length in the differentiation zone. Data are presented as means ± sd (sample size is indicated by gray circles) and, for each genotype, the statistical differences between treatments were analyzed using a two-way ANOVA followed by Šídák's multiple comparisons test (***$P < 0.001$; ****$P < 0.0001$). **e** Confocal microscopy

images showing the quiescent center (QC) of the roots shown in (**a**). Scale = 20 μm. **f** Pictures showing the lengths of the cells in the root differentiation zone. Scale = 25 μm. In (**e**) and (**f**), the root cell walls were stained with Calcofluor White and visualized using confocal microscopy. Analysis was repeated on a set of ten individual root meristems with similar results. **g** Visualization of the iron distribution at the root tip, obtained using Perls-DAB staining of plants grown in +Pi and −Pi. Scale = 200 μm. Data from an independent experiment is shown in Supplementary Fig. 17. Source data for (**b**–**d**), including *n* and *p* values, are provided as a Source data file.

phenotype, no significant difference was found in the meristem size or cell length in the elongation zone between Col-0 and the mutant lines (Fig. 3b). In −Pi, the macroscopic analysis of the root phenotypes revealed that the only fully epistatic interaction was between *lpr1* and

*crr-1*, as the primary root length of *crr-1 lpr1* was not different from the *lpr1* single mutant (Fig. 3a). In addition, *crr-1* showed partial epistasis with *almt1* for primary root length, with the *crr-1 almt1* double mutant producing a root length significantly lower than *almt1* but longer than

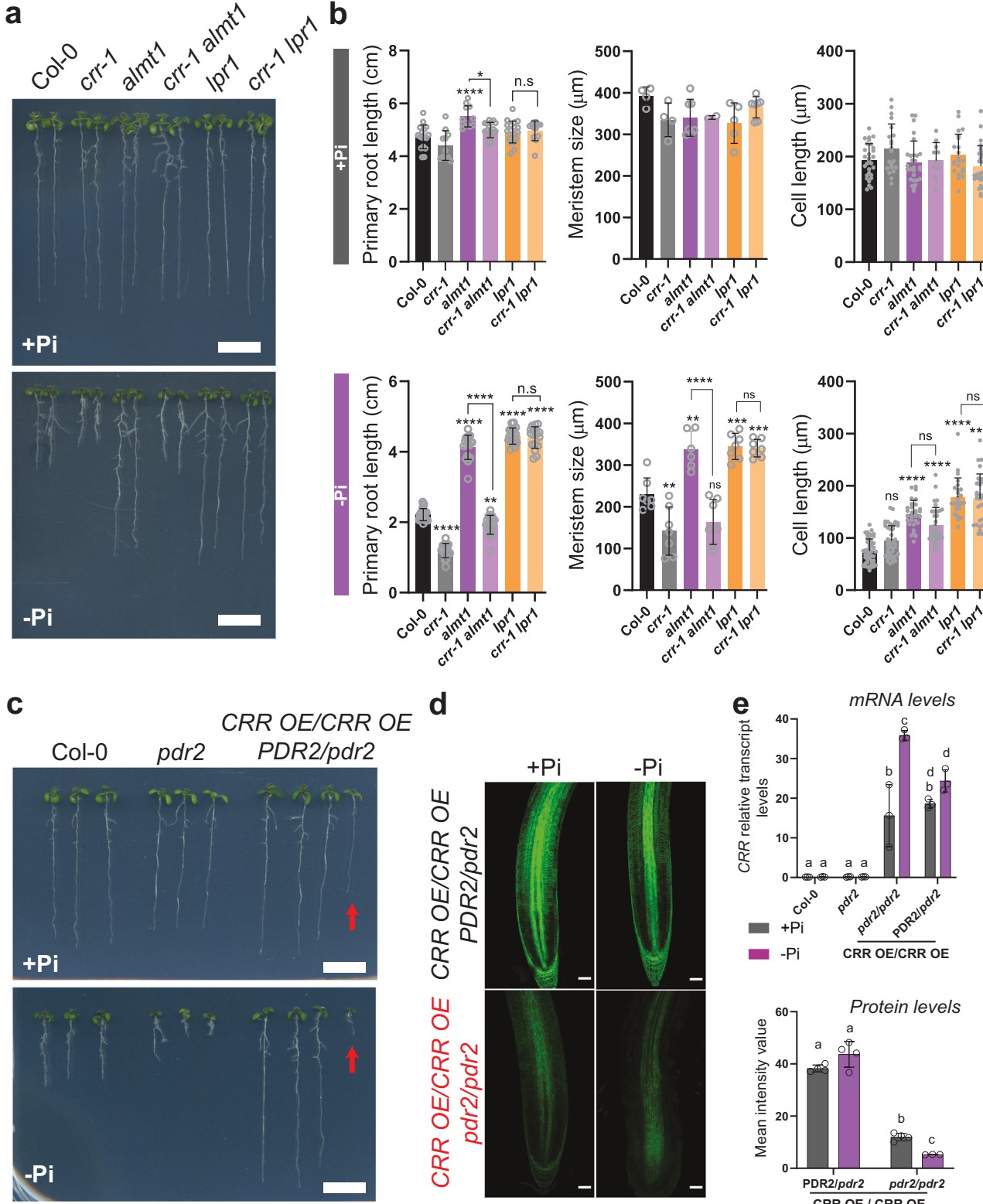

crr-1. A microscopic analysis of the roots revealed that the meristem size and cell length in the elongation zone of the *lpr1* roots was not affected by the introgression of *crr-1*. In the case of *almt1*, the introgression of *crr-1* significantly reduced meristem division to a level similar to the *crr-1* single mutant but did not influence cell elongation (Fig. 3b). These results further support the idea that CRR uncouples the meristem exhaustion process from cell elongation inhibition and explains why there is no complete epistasis between *crr* and *almt1*; *crr*

is epistatic to *altm1* on its effect on meristem size under −Pi but has no effect on cell elongation (Fig. 3b).

To examine the genetic interaction between *CRR* and *PDR2*, the transgenic line CRR OE overexpressing *CRR-GFP* was crossed with the *pdr2-1* mutant. The F₃ progeny from a line homozygous for the *CRR-GFP* transgene and heterozygous for *pdr2-1* were analyzed. The inspection of roots under −Pi showed that the F₃ seedlings developed either a long-root phenotype resembling CRR OE or a hypersensitive

**Fig. 3 | Genetic interactions between *CRR* and *ALMT1, LPR1* or *PDR2*. a** Primary root growth responses of plants grown in +Pi and −Pi media for 7 days. Scale = 1 cm. **b** Quantification of the primary root length, meristem size, and cell length in the differentiation zone of the roots of the different genotypes shown in (**a**). Data are presented as means ± sd (sample size is indicated by circles). The statistical differences were analyzed using a one-way ANOVA followed by Tukey's multiple comparisons test. Significant differences against Col-0 are indicated by asterisks. In addition, selected comparisons and their statistical significance are indicated with brackets (****$P < 0.0001$; ***$P < 0.001$; **$P < 0.01$; *$P < 0.05$). **c** CRR OE and *pdr2-1* were crossed, and F$_3$ seeds from a line homozygous for the CRR OE cassette but heterozygous for the *pdr2-1* allele were selected for analysis and compared with Col-0 and the parental *pdr2-1*. One-quarter of the segregating population showed a short-root phenotype and were genotyped as homozygous *pdr2-1* (representative seedling highlighted by a red arrow), while all plants with a long-root phenotype were genotyped as either *PDR2/PDR2* or *PDR2/pdr2-1*. **d** Confocal microscopy images showing *CRR-GFP* expression levels in the *PDR2/pdr2-1* or *pdr2-1/pdr2-1* backgrounds for plants grown in +Pi and −Pi media. Scale = 50 μm. **e** Quantification of *CRR* transcript levels using qRT-PCR (upper panel) and CRR-GFP protein levels (lower panel) based on confocal microscopy images. The data represents the mean ± sd (sample size is indicated by circles). The statistical differences were analyzed using a two-way ANOVA followed by Tukey's multiple comparisons test. Different letters indicate significant differences ($P < 0.05$). Source data for (**b**) and (**e**), including n and p values, are provided as a Source data file.

short-root phenotype similar to *pdr2-1* in a ratio of 3:1 (Fig. 3c). Genotyping showed that all seedlings with a short-root phenotype were *pdr2-1/pdr2-1* homozygous, while all seedlings with a long-root phenotype were *PDR2/PDR2* or *PDR2/pdr2-1*, thus demonstrating that the *pdr2-1* mutation is epistatic to CRR OE. PDR2 is localized in the ER, and the *pdr2-1* mutant is unable to maintain the integrity of the ER marker HDEL-GFP (Ticconi et al.[32]); therefore, we examined *CRR-GFP* expression levels in the *pdr2-1* mutant background. The results showed that the CRR-GFP protein levels were strongly reduced when introgressed in *pdr2-1*, while the mRNA levels remained unaffected, suggesting that a functional PDR2 is required for CRR protein stability (Fig. 3d, e).

### The *crr* short-root phenotype under −Pi is enhanced by factors contributing to Fenton reactions

Fe-mediated Fenton reactions generating ROS have been implicated in the inhibition of primary root elongation under Pi deficiency, and such reactions are enhanced by low pH and $H_2O_2$ levels[42]. Primary root growth of Col-0, *crr-1*, and CRR OE grown in +Pi and −Pi media with different pH (5.2 to 6.0) (Supplementary Fig. 18a) or $H_2O_2$ contents (Supplementary Fig. 18b) was thus assessed. In +Pi, medium pH was found to result in similar root phenotypes in all the genotypes tested. In −Pi, however, the short-root phenotype of *crr-1* was greatly enhanced at low pH, while at higher pH, the *crr-1* root length was not different from Col-0 (Supplementary Fig. 18a). For all pH levels tested under −Pi, the primary root of the CRR OE line was longer than that of Col-0. Similarly, the addition of $H_2O_2$ in the +Pi medium had little influence on primary root growth, while it exacerbated *crr-1* root growth inhibition under −Pi (Supplementary Fig. 18b). These results support the hypothesis that the short-root phenotype of *crr-1* observed under −Pi depends on the amount of Fe accumulated at the SCN that can undergo Fenton reactions. An effect of pH on *ALMT1* expression could also contribute to the *crr-1* primary root growth under −Pi condition[35].

### CRR has ascorbate-dependent ferric reductase activity affecting root redox homeostasis

The potential implication of CRR acting in $Fe^{3+}$ reduction was first tested by measuring ferric reductase activity in the roots of the CRR OE line using Fe(III)-malate, Fe(III)-EDTA or FeCN ($[Fe(III)(CN)6]^{3-}$) as substrate. As a positive control, a line overexpressing the transcription factor bHLH39 (39OE) resulting in increased *FRO2* expression and high ferric reductase activity was used[58]. Analysis showed that roots overexpressing *CRR* had a higher Fe(III)-malate, Fe(III)-EDTA, and FeCN reductase activity than Col-0 (Fig. 4a). While CRR OE and the 39OE lines showed similar increase in reductase activity when using FeCN, line 39OE showed relatively greater reductase activity with Fe(III)-EDTA and Fe(III)-malate compared to CRR OE. These findings suggest that CRR and the canonical FRO2 ferric reductase involved in Fe homeostasis have different affinities for various Fe−chelator complexes and might be involved in different processes. In agreement with this hypothesis, the primary root lengths of *irt1* and *fro2* grown in −Pi were similar to that of Col-0, indicating that neither FRO2 nor its associated $Fe^{2+}$ transporter IRT1 influence the primary root growth response in −Pi conditions (Supplementary Fig. 19). As previously reported, the line 39OX showed strongly reduced root growth in +Pi which was also evident under −Pi[58]. Previous analyses have shown *FRO2* expression in the root hairs and epidermal cells of the mature root tissues but not in the root meristematic region[59,60]. Analysis of single-cell transcriptomic data of root tips shows a lack of *FRO2* expression in cells forming the root meristem (e.g., SNC) (Supplementary Fig. 20).

Analysis of ferric reductase activity in *crr-1* and *crr-1 hyp1* roots showed a statistically significant small reduction in reductase activity only with Fe(III)-malate and Fe(III)-EDTA in the *crr-1 hyp1* double mutant but not in the *crr-1* single mutant (Fig. 4a). The relatively narrow expression pattern of *CRR* in the root meristem likely impact the detection of ferrireductase activity in roots of the single *crr-1* mutant.

The ferric reductase activity of CRR was also assessed via its expression in the mutant *Saccharomyces cerevisiae* strain *Δfre1 Δfre2* lacking the two main endogenous ferric reductases, FRE1 and FRE2. As shown in Fig. 4b and Supplementary Fig. 21, *CRR* expression did not enhance ferric reductase activity in the *Δfre1 Δfre2* strain using either Fe(III)-malate, Fe(III)-EDTA or FeCN. Some CYB561 proteins in animals and plants use ascorbate as an electron donor[48]; however, *S. cerevisiae* is unable to biosynthesize ascorbate unless a chemical precursor like L-galactono-lactone is added to the medium[61]. When this precursor was added to the yeast growth medium, the expression of *CRR* in *Δfre1 Δfre2* resulted in a strong increase in ferric reductase activity with FeCN, and weaker increase with Fe(III)-malate compared with the empty vector control (Fig. 4b and Supplementary Fig. 21). No significant differences were observed when Fe(III)-EDTA was used as substrate (Supplementary Fig. 21). This observation of stronger ferric reductase activity of yeast-expressed CRR with FeCN compared to other substrates is consistent with previous reports analyzing the ferrireductase activity of distinct cytochromes b561 containing proteins in heterologous systems[62,63]. Taken together, these results show that CRR can act as an ascorbate-dependent ferric reductase sensitive to different Fe chelation forms.

To assess how the ferric reductase activity of CRR would affect gene expression in roots, in particular under −Pi where it displays a strong phenotype, a transcriptomic analysis of Col-0, *crr-1*, and CRR OE roots grown under +Pi and −Pi was performed (Fig. 4c). Three biological replicates per condition were sequenced, obtaining more than 25 million reads per library (Supplementary Data 2 and 3). A dendogram and a principal component analysis (PCA) showed that in +Pi, *crr-1* and CRR OE samples cluster together with Col-0, indicating that CRR expression levels barely impacts the root transcriptome in control conditions. In −Pi, *crr-1* and CRR OE diverged from the WT, suggesting that *CRR* expression levels, either in ectopic cells or in cell normally expressing the endogenous *CRR*, significantly perturbs root gene expression under low-Pi conditions (Supplementary Figs. 22 and 23). In agreement with this, an analysis of the differentially expressed genes (DEGs) revealed that in +Pi only 6 genes (3 upregulated and 3 downregulated) were differentially expressed in *crr-1* and 22 (10 upregulated and 12 downregulated) in CRR OE, when compared to Col-0. On the

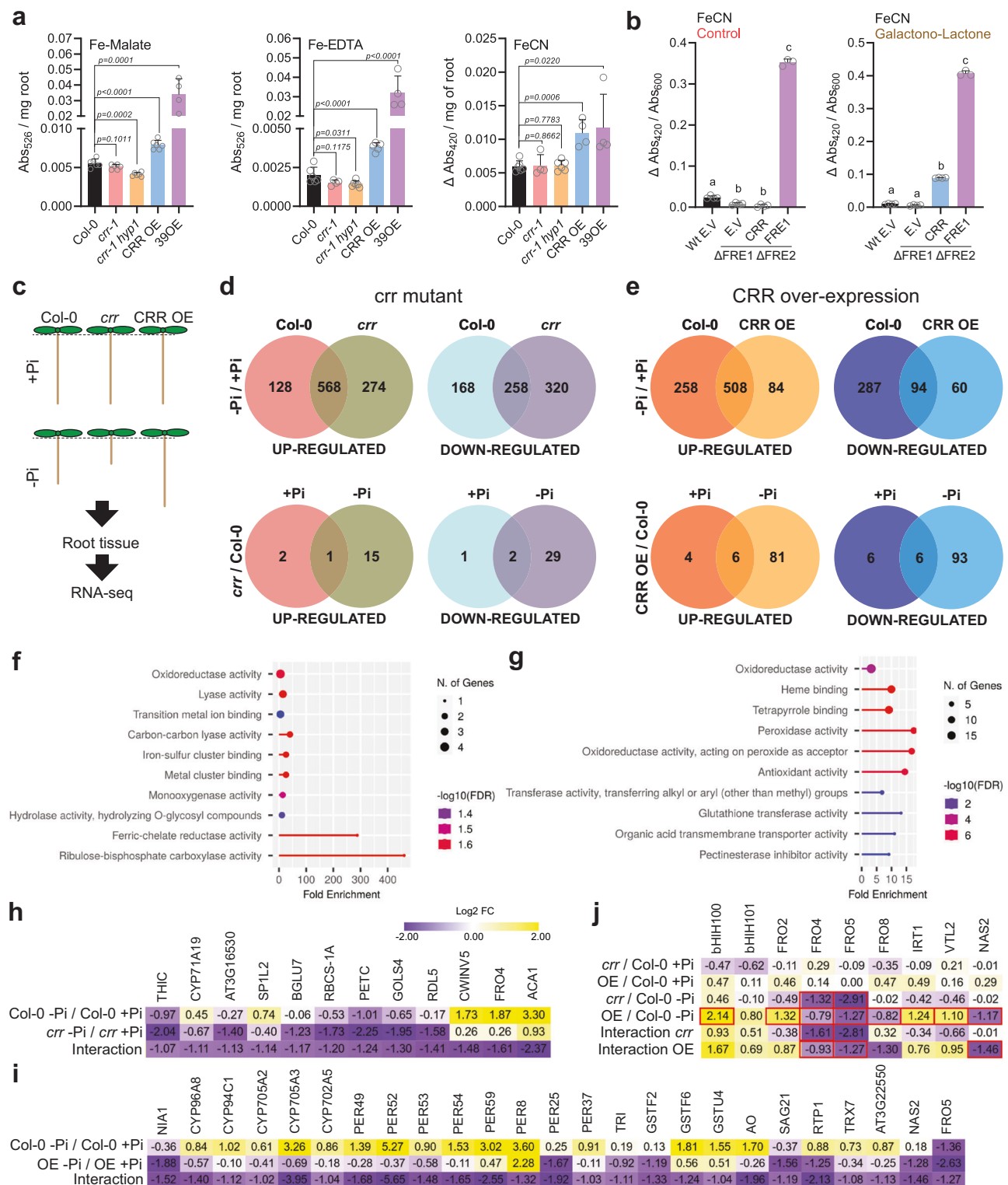

other hand, in −Pi there were 47 DEGs (16 upregulated and 31 down-regulated) in *crr-1* and 186 DEGs in CRR OE (87 upregulated and 99 downregulated, respectively) (Fig. 4d, Supplementary Figs. 24a, 25a, and Supplementary Data 4–7).

An analysis of the DEGs between the −Pi/+Pi treatments revealed a large overlap among the upregulated genes in Col-0 and *crr-1* mutant or the *CRR* overexpressing line. A GO analysis of this group showed that the main enriched terms were related to Pi starvation, Pi homeostasis, and Pi metabolism, suggesting that even though the primary root growth is different in these lines under −Pi, the systemic

Pi deficiency signaling pathway was activated in a similar way (Supplementary Figs. 24b and 25b). A detailed inspection of the canonical Pi starvation–responsive genes (e.g., *IPS1*, *SPX1*, *MGD2*, *PHT1;2*, and *PAP17*) indicated that all were upregulated to similar levels independently of the genotype (Supplementary Figs. 24c and 25c). Pi quantification in the roots and shoots showed that Pi levels in Col-0, *crr-1*, and CRR OE were not significantly different, with a similar decrease in roots grown under −Pi conditions (Supplementary Fig. 25d). Overall, these data indicate that the *crr-1* and CRR OE phenotypes are restricted to primary root development under low-Pi

**Fig. 4 | CRR has ferric reductase activity and its expression levels impact the root transcriptome. a** Ferric reductase activity in roots. FeCN reduction was measured as changes in absorbance at 420 nm and Fe-EDTA/Malate reduction was measured by the formation of ferrozine-$Fe^{2+}$ complexes which absorb at 526 nm. The concentration of Fe-Malate and Fe-EDTA was 100 μM and 500 μM for FeCN. Statistical differences against Col-0 were analyzed using an unpaired two-tailed t-test. **b** The *Saccharomyces cerevisiae* strain S288C or the FRE1 and FRE2 mutant (*Δfre1 Δfre2*) were transformed with a plasmid driving the expression of *CRR* or *FRE1* under the control of the *GPD* promotor or the empty vector. The strains were grown in synthetic defined (SD) media or SD supplemented with the ascorbate precursor galactono1,4-lactone. FeCN (500 μM) was used as substrate and iron reduction was quantified as in (**a**). Statistical differences were analyzed using a one-way ANOVA followed by Tukey's multiple comparisons test. In (**a**) and (**b**), data represent the means of different biological replicates ± sd (***$P < 0.001$; **$P < 0.01$; *$P < 0.05$). **c** Schematic representation of the RNA-seq experimental design.

**d**, **e** Venn diagrams showing the differentially expressed genes (DEGs) of the treatment (−Pi/+Pi) or genotype responses for *crr-1* and CRR OE, respectively. **f**, **g** Gene ontology (GO) enrichment analysis of the downregulated DEGs found in the interaction between genotype and treatment defined for *crr-1* as ((crr-1 −Pi/Col-0 −Pi)/(crr-1 +Pi/Col-0 +Pi)) and for CRR OE as ((CRR OE −Pi/Col-0 −Pi)/(CRR OE +Pi/Col-0 +Pi)), respectively. The graph shows the top 10 significantly enriched 'molecular function' GO terms. For each category, the fold enrichment, the number of genes, and the false discovery rate (FDR) are shown. **h** Expression pattern of the downregulated genes found in the interaction between *crr-1*, Col-0, and treatment. **i** Expression pattern of DEGs associated with redox process functions found among the downregulated genes in the interaction between CRR OE, Col-0, and treatment. **j** Expression pattern of iron-related genes. Red boxes indicate statistically significant differences. Numerical values in (**h**–**j**) represent log2 fold-change. Exact *n* and *p* values are provided in the Source data file.

conditions and that the Pi acquisition and Pi deficiency responses are not affected.

We then analyzed the DEGs in the interaction between genotypes and treatments ((*crr-1* or CRR OE −Pi/Col-0 −Pi)/(*crr-1* or CRR OE +Pi/Col-0 +Pi)), which highlights how each genotype differs from Col-0 in their response to −Pi treatments. A total of 25 (13 up- and 12 down-regulated) and 190 (98 were upregulated and 92 downregulated) DEGs were identified between Col-0 and *crr-1* or CRR OE, respectively (Supplementary Data 4–7). Interestingly, a gene ontology (GO) analysis of the genes downregulated in *crr-1* and CRR OE showed a strong enrichment in categories related to redox homeostasis (e.g., oxidoreductase activity for *crr-1* and CRR OE, peroxidase activity and antioxidant activity for CRR OE) and Fe binding (iron-sulfur cluster for *crr-1*, heme and tetrapyrrole binding for CRR OE) (Fig. 4f, g).

Detailed analysis showed that *FRO4* was differentially expressed in *crr-1* (Fig. 4h). FRO4, together with FRO5, have been related to copper (Cu) reduction and uptake[64]. This observation might either indicate that CRR is directly involved in Cu homeostasis (supported by the described Cu-reductase activity of HYP1 reported in the accompanying work of Maniero et al.[53], or that its ferrireductase activity indirectly affect Cu homeostasis as a consequence of the crosstalk between Cu, Fe, and Pi signaling pathways[64,65]. Also of interest is the downregulation of GALACTINOL SYNTHASE 4 (GOLS4), a member of the galactinol synthase protein family implicated in plant responses to different abiotic stresses via ROS-mediated oxidative damage[66–68] (Fig. 4h).

Analysis of the genes downregulated in the CRR OE line compared to Col-0 (interaction genotype x treatment) showed that several cytochromes p450, peroxidases, and glutathione *S*-transferases were differentially expressed (Fig. 4i). This observation suggests that CRR ectopic expression modifies roots ROS levels through a modification of cellular iron homeostasis. Indeed, NAS2, which is required for the production of nicotianamine, a key compound involved in iron distribution and transport, was also downregulated[69]. This finding prompted us to inspect the expression of genes directly involved in Fe homeostasis. The *bHLH100* gene, encoding a key regulator of the Fe deficiency responses, was downregulated in the −Pi Col-0 roots; however, this inhibition was attenuated in the CRR OE line. Conversely, the upregulation of *FRO8* in the −Pi Col-0 roots was attenuated in −Pi CRR OE (Fig. 4j). FRO8 has not been functionally characterized but it is believed to reduce iron in the mitochondrial membrane[70]. In addition, a number of Fe-related genes were mildly deregulated in CRR OE. Thus, the *bHLH101*, *FRO2* and *IRT1* genes that are normally activated by Fe deficiency, as well as the *VTL2* gene typically repressed under similar condition, were all slightly upregulated in CRR OE. Interestingly, FRO4 was found as differentially expressed both in *crr-1* and CRR OE roots (Fig. 4j). Finally, a GO analysis based on cellular component terms revealed that most of the DEGs in the interaction between genotypes and treatments were localized at the cell wall and/or in the apoplastic space (Supplementary Figs. 24d and 26).

We next analyzed ROS production in the root apical meristem using the general ROS probe carboxy-H2CDFDA (Supplementary Fig. 27). As previously reported[30], Col-0 roots show enhanced production of ROS in the meristem under Pi deficiency, particularly in the SCN. This response is exacerbated in *pdr2* which is consistent with the higher accumulation of iron in the root meristem. Interestingly, under −Pi, *crr-1* and *crr-1 hyp1* showed an increase of carboxy-H2CDFDA signal as compared to Col-0, not only in the SCN but also all along the meristem division zone. This pattern of ROS production was stronger in *crr-1 hyp1* double mutant, explaining the shorter root phenotype compared to the *crr-1* single mutant, strengthening the correlation between iron accumulation, ROS, and root growth inhibition. In line with this, the carboxy-H2CDFDA staining in CRR OE meristems did not show any obvious difference between +Pi and −Pi (Supplementary Fig. 27). Taken together, these results indicate that CRR activity affects the redox status of the root apoplastic space as a consequence of changes in the amount of Fe accumulated in the apoplast, explaining the root growth phenotypes in −Pi.

### *CRR* expression levels affect *A. thaliana* tolerance to Fe stress

The findings that CRR has ferric reductase activity, that its overexpression causes the deregulation of Fe-homeostasis and redox-related genes, and that its expression level determines the amount of Fe accumulated at the root tip under −Pi conditions led us to speculate that CRR (and possibly other members of the CYBDOM family) might have a broader biological functions in Fe homeostasis. Because the *crr-1*, *crr-1 hyp1*, and CRR OE phenotypes are associated with conditions in which Col-0 roots hyperaccumulate Fe, we investigated the response of these lines to high Fe levels.

Col-0, *crr-1*, CRR OE, and *crr-1 hyp1* were challenged to high amounts of Fe supplied as Fe(III)-EDTA for 7 days from germination. Remarkably, CRR OE was hypersensitive to Fe stress, showing a mild to strong reduction in the fresh weight of its seedlings in media containing 400 to 600 μM Fe(III)-EDTA, respectively (Fig. 5a–c). Conversely, the *crr-1* and *crr-1 hyp1* mutants were more tolerant of the high-Fe stress in terms of the biomass generated. Similar results were obtained when seeds were sown in −Pi media with high Fe, with the exception that the Fe toxicity symptoms appeared at lower concentrations (350 μM Fe(III)-EDTA), most likely because the removal of Pi enhances Fe accessibility via a reduction in the formation of insoluble Pi−Fe complexes (Supplementary Fig. 28a, b). Similar contrasting growth phenotypes were observed for the CRR OE and *crr-1 hyp1* plants grown for 7 days in a control basal MS medium and transferred into a medium with an additional 600 μM Fe(III)-EDTA for 4 days (Supplementary Fig. 28c, d).

The phenotypes associated with a high-Fe supply were also analyzed for the CRR OE, *crr-1* and *crr-1 hyp1* grown under phototrophic growth conditions. In a clay-based substrate, Col-0 germination and growth were negatively affected when supplemented with a

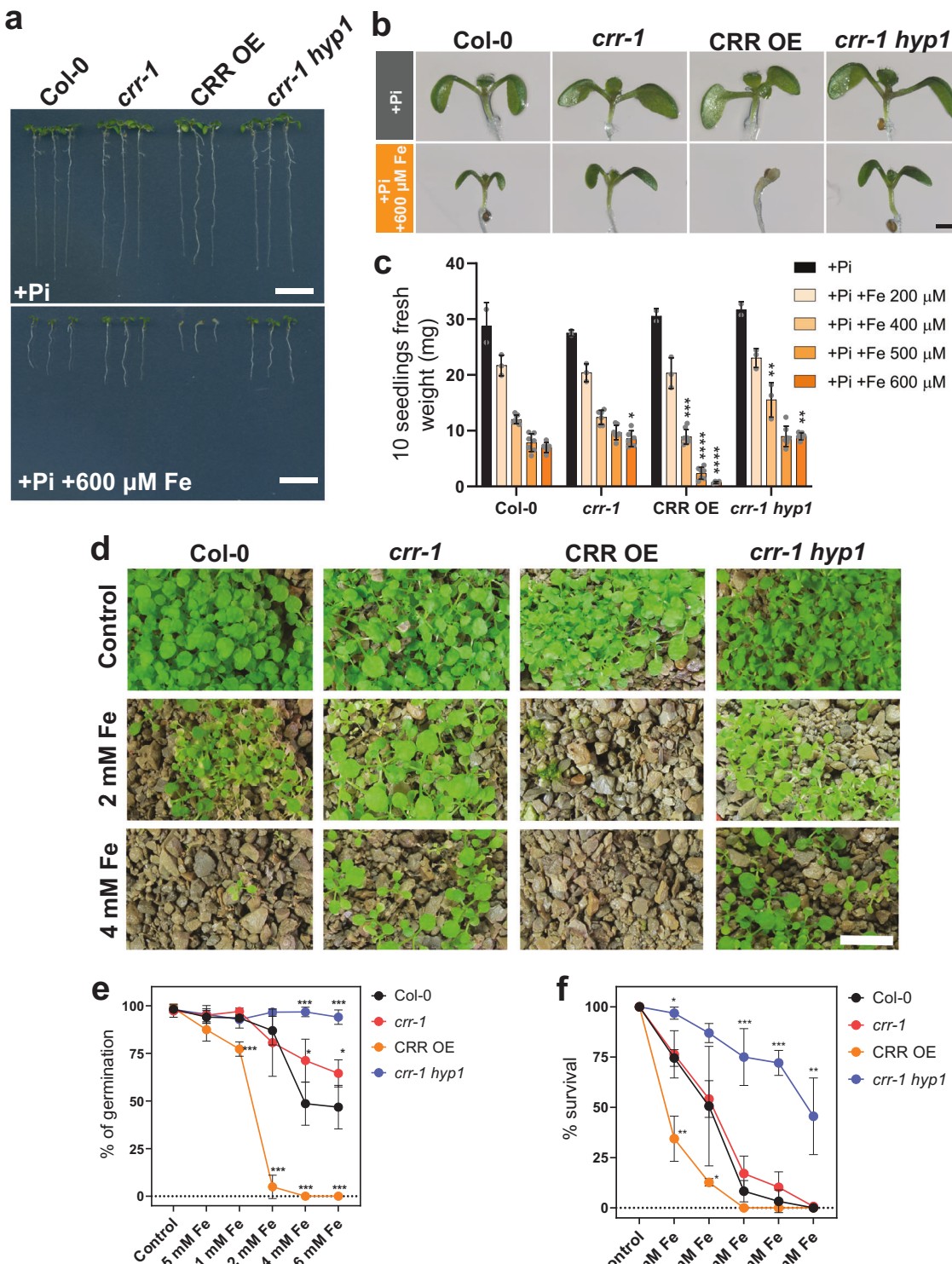

**Fig. 5 | CRR is involved in iron toxicity tolerance. a** Phenotypes of Col-0, *crr-1*, CRR OE, and the double mutant *crr-1 hyp1* grown for seven days in the +Pi medium with or without an additional 600 µM Fe provided as Fe-EDTA. Scale = 1 cm. **b** Magnification showing the shoot phenotype of the seedlings photographed in (**a**). Scale = 0.5 mm. **c** Effect of increasing the concentration of Fe-EDTA on Arabidopsis seedling growth, determined by measuring the weight of 10 seedlings after 7 days of growth. **d** Phenotypes of 15-day-old seedlings sown in a clay-based substrate watered with increasing concentrations of Fe-EDTA. Scale = 1 cm. **e** Percentage of seeds that germinated in the clay-based substrate watered with increasing concentrations of Fe-EDTA, measured after five days. **f** For the plants in (**e**) that had germinated by day 5, the percentage of plants that remained green and continued to develop for an additional 10 days was quantified. In (**e**) and (**f**) $n = 3$ in all cases. In (**c**), (**e**), and (**f**), the data are the means ± sd and sample size is indicated by dots. The statistical differences within each condition were analyzed using a one-way ANOVA followed by Dunnett's multiple comparisons test. Significant differences from Col-0 are indicated (****$P < 0.0001$; ***$P < 0.001$; **$P < 0.01$; *$P < 0.05$). Source data for (**c**), (**e**) and (**f**), including $n$ and $p$ values, are provided as a Source data file.

2 mM Fe(III)-EDTA solution, while CRR OE was unable to germinate under the same conditions (Fig. 5d, e). Conversely, both the *crr-1* and *crr-1 hyp1* mutants were clearly more tolerant of Fe toxicity, showing higher levels of germination compared with Col-0 at Fe(III)-EDTA concentrations of 4–6 mM (Fig. 5d, e). Furthermore, of the seeds that germinated, the *crr-1 hyp1* mutant showed the highest level of survival after 15 days of stress, while CRR OE had the lowest survival rate (Fig. 5f). The germination and survival rates of the *crr-1 hyp1* double mutant under high Fe were both higher than those of the single *crr-1* mutant (Fig. 5e, f). Similar results were obtained when the experiments were performed with plants grown in peat-based soil (Supplementary Fig. 28e–g).

## CRR impacts Fe accumulation in the shoots

To understand the physiological mechanisms behind the high Fe-associated phenotypes of the *crr-1/crr-1 hyp1* mutants and CRR OE, the Fe distribution in the roots of seedlings grown in agar plates was examined. Perls-DAB staining of roots showed that *crr-1* and *crr-1 hyp1* accumulated more Fe at the root tip than Col-0 under Fe stress conditions, whereas CRR OE accumulated less (Supplementary Fig. 29). Perls staining without the DAB intensification of roots grown in +Pi medium with basal level of Fe revealed that while Col-0, *crr-1*, and *crr-1 hyp1* accumulated Fe at the QC and the root cap with only faint staining in the surrounding initials under control conditions, CRR OE showed similar staining in the QC and initials, but only very weak Fe accumulation in the root cap (Fig. 6a and Supplementary Fig. 30). In the elongation and differentiation zones, Fe deposition was below the detection limits for all lines. In the +Pi medium supplemented with 350 μM Fe(III)-EDTA, Col-0 showed Fe staining mainly in the QC and weakly in the surrounding initials, a pattern similar to CRR OE under control conditions. In contrast to Col-0, *crr-1* and *crr-1 hyp1* displayed higher levels of Fe at the root meristem, including in the QC and initials. There was an increase in Fe staining intensity in the elongation and differentiation zones of the *crr-1* single and *crr-1 hyp1* double mutants. In CRR OE, Fe was undetected in the root meristem and only a weak signal was obtained in the elongation and differentiation zones (Fig. 6a and Supplementary Fig. 30).

Our analysis of Fe accumulation in the cotyledons using Perls-DAB staining showed that, when grown in a medium containing 350 μM Fe(III)-EDTA, CRR OE clearly accumulated more Fe whereas the difference between Col-0 and the *crr-1* or *crr-1 hyp1* was not reliably detectable (Fig. 6b and Supplementary Fig. 31). The possibility that Fe accumulation in the shoots of CRR OE was associated with increased ROS production was tested using carboxy-H2DCFDA. As shown in Fig. 6c, the CRR OE shoots had higher levels of ROS when grown under both control conditions and with a 350 μM Fe(III)-EDTA supply. To test if iron accumulation was also modified in true leaves, we performed a Perls-DAB staining of shoots from Arabidopsis lines for 11 days under +Pi conditions or a milder iron stress (+Pi with 200 μM Fe(III)-EDTA) (Supplementary Fig. 32). The results showed that compared to Col-0, CRR OE hyperaccumulated Fe in true leaves whereas Fe staining was weaker in *crr-1 hyp1* double mutant.

Overall, an analysis of the Fe staining in both the roots and shoots revealed an antagonistic pattern between CRR OE and the *crr-1/crr-1 hyp1* mutants, with CRR OE accumulating more Fe in the shoots than the roots, resulting in ROS generation in the shoots. By contrast, the *crr-1* and *crr-1 hyp1* mutants accumulated more Fe in the roots and, for *crr-1 hyp1*, less in the shoots, suggesting that Fe reduction by CRR (and HYP1) may be associated with its transport from the roots to the shoots.

To further elucidate how CRR could modulate Fe transport, CRR OE and mutant lines grown for 7 days on a control medium were transferred into a medium containing 600 μM Fe-EDTA and supplemented with $^{55}$Fe for 4 days before measuring the amount of $^{55}$Fe in the roots and shoots. Consistent with the Perls-DAB staining, CRR OE contained a higher amount of $^{55}$Fe per milligram of tissue than Col-0 in the shoots and a lower amount in the roots, with a two-fold greater level of root-to-shoot $^{55}$Fe transfer than in the WT (Fig. 6d). For *crr-1* and *crr-1 hyp1*, no significant difference was measured relative to Col-0. The apparent discrepancy between the Perls-DAB staining and $^{55}$Fe uptake/transfer experiment for the *crr-1 hyp1* double mutant may indicate that Fe accessibility or compartmentation in leaves may be more affected than the transport dynamic per se. Altogether, these results suggest that *CRR* overexpression results in the altered accumulation of Fe in the shoots, affecting plant fitness under high Fe concentrations.

Finally, we analyzed whether the phenotype of CRR OE on shoot growth and Fe accumulation was largely dependent on CRR overexpression in roots, shoots or both. To assess this, we performed homo- and hetero-grafts using Col-0 and CRR OE seedlings. As expected, after 6 days under high iron stress, CRR OE homo-grafts showed a reduced shoot biomass and higher Fe staining by Perls-DAB compared to Col-0 (Supplementary Fig. 33). Interestingly, either when CRR OE was used as scion or rootstock, shoot growth and Fe staining in the reciprocal hetero-grafts were similar to the self-grafted CRR OE. These data indicate that CRR overexpression in both shoots and roots contribute to the high shoot Fe and deleterious growth observed under high external Fe supply.

## Discussion

Studies on the response of primary roots to Pi deficiency in *A. thaliana* have established a link between the presence of $Fe^{3+}$ in the root tip apoplast with ROS production and the reduction in meristematic cell division and cell elongation[30,35–38,40]. The central role of the ferroxidase *LPR1* in these phenotypes indicates that, despite the presence of adequate amount of $Fe^{3+}$ in the external medium, LPR1 must oxidize $Fe^{2+}$ de novo in the root meristem apoplast in order to mediate reduction of primary root growth[30,40]. This raises a number of questions, such as the potential presence of specific apoplastic (sub)domains with limited access to $Fe^{+3}$ as well as how $Fe^{2+}$ would be generated in the root meristem for LPR1 activity. In *A. thaliana*, FRO2 is the primary root ferric reductase involved in Fe import into plants[71]. FRO2 is localized to the plasma membrane and primarily expressed in the root epidermis and root hairs[59,60,72]. Although FRO2 paralogues are present in other subcellular membranes (e.g., FRO3/8 and FRO7 in mitochondria and plastids, respectively), FRO2 remains the only FRO family member known to be involved in Fe reduction in the apoplast[73]. While some coumarins secreted under Fe-deficient conditions at high external pH exhibit intrinsic Fe-reducing properties, their role in Fe acquisition at neutral and more acid pH levels appears negligible[74–76]. Furthermore, even under alkaline conditions, Fe mobilized by coumarins requires the action of FRO2 for its acquisition[77]. Under phosphate deficiency, *FRO2* and *IRT1* transcription is strongly downregulated and unlikely to play a major role in Fe reduction and acquisition, at least in the root meristem[13,15–17]. The primary roots of both the *fro2* and *irt1* mutants show WT-like responses to Pi deficiency, as shown in this work and that of ref. 40, indicating that neither participate in the response of primary roots to Pi deficiency. Although apoplastic ascorbic acid has been shown to participate in $Fe^{3+}$ reduction for $Fe^{2+}$ import into developing *A. thaliana* seeds, it is unknown whether ascorbic acid contributes to $Fe^{3+}$ reduction in the roots[78].

Biochemically characterized CYB561s have shown the ability to use ascorbate as the principal electron donor to generate transmembrane electron transport to reduce either apoplastic monodehydroascorbate or various ferric-chelates[48]. A CYB561 found in the mouse duodenal mucosa cell membrane transfers electrons from cytoplasmic ascorbate to extracellular $Fe^{3+}$-chelates, enabling the generation of $Fe^{2+}$ ions that are then acquired by $Fe^{2+}$ transporters[79–81]. The characterization of some plant CYB561s showed that they also use ascorbate to generate transmembrane electron transfers to

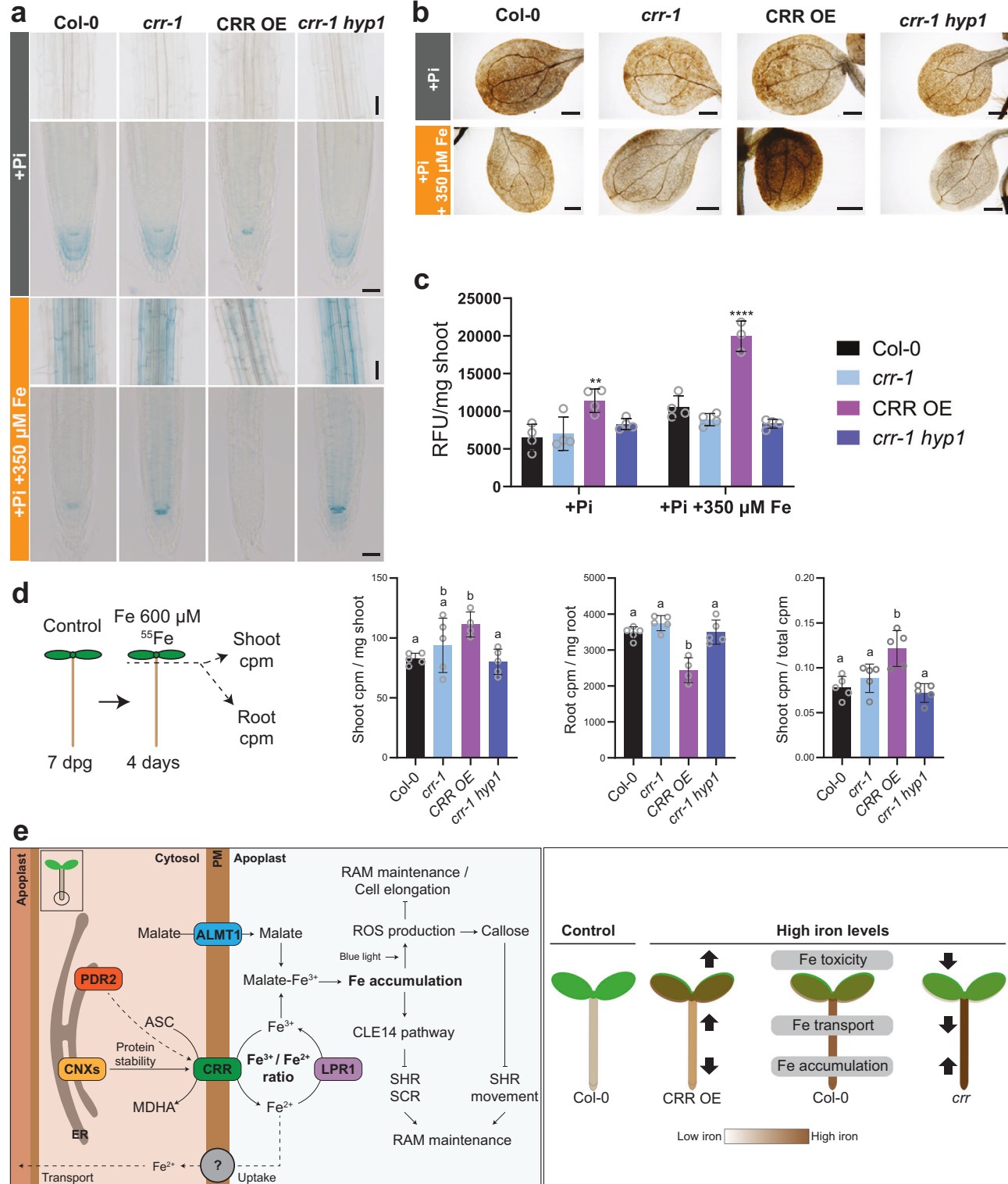

monohydroascorbate or $Fe^{3+}$-chelates when expressed in heterologous systems, such as *Pichia pastoris*[62,82,83]. Despite these insights, the in vivo relevance of CYB561 proteins to either $Fe^{3+}$ reduction and iron homeostasis or ascorbate pools has not been demonstrated in plants.

The DOMON domain is commonly present as an extracellular domain in membrane-anchored proteins and is often fused to domains other than CYB561 that are involved in redox reactions, such as monooxygenase, cytochrome c, or FAD/NAD-binding

oxidoreductase[84]. While only a single CYBDOM protein is found in human, mouse, and *Drosophila*, the CYBDOM protein family is greatly expanded in plants. The DOMON domain of several proteins has been found to bind heme in bacteria and fungi[49,85]. In plants, the AIR12 protein in soybean (*Glycine max*) and *A. thaliana* has a single DOMON domain with a GPI anchor and has been shown to bind heme that is fully reduced by ascorbate[46]. Although the mode of action of AIR12 is poorly defined, its expression has been associated with superoxide

**Fig. 6 | CRR impacts iron accumulation in the roots and its transport to the shoots. a** Phenotype of Col-0, *crr-1*, CRR OE, and *crr-1 hyp1* plants grown for seven days in the +Pi medium as a control, or in the +Pi medium supplemented with 350 μM Fe-EDTA. The roots were subjected to Perls staining to detect iron as a blue precipitate. Scale = 350 μM. **b** Perls-DAB staining showing Fe accumulation patterns in the cotyledons. Plants were grown as in (**a**), and the shoots were subjected to Perls staining followed by a DAB intensification step. Scale = 500 μm. **c** ROS in the shoots of plants grown as in (**a**), quantified using the fluorescence probe Carboxy-H2DCFDA and expressed as relative florescence units (RFU) per mg of tissue. The statistical differences within each condition were analyzed using a one-way ANOVA followed by Dunnett's multiple comparisons test. Significant differences from Col-0 are indicated (****$P < 0.0001$; **$P < 0.01$) and the sample size by circles.

**d** Schematic representation of the experimental design (left). Plants were grown for seven days in +Pi control conditions and transferred onto a +Pi medium containing 600 μM Fe-EDTA and 1.7 μCi per ml of $^{55}$Fe. After four days, the shoots and roots were collected separately, extensively washed, and the amount of $^{55}$Fe in the tissue was quantified as counts per million (cpm). Results are expressed as shoot cpm per mg of shoot, root cpm per mg of root, or the ratio shoot cpm over total plant cpm. The statistical differences were analyzed using a one-way ANOVA followed by Tukey's multiple comparisons test. Different letters indicate significant differences ($P < 0.05$). In (**c**) and (**d**), the data are the means ± sd and each individual value is indicated by a circle. **e** Working model of CRR function under phosphate deficiency and iron stress. The model is explained in the Discussion. Source data for (**c**) and (**d**), including *n* and *p* values, are provided as a Source data file.

generation at the plasma membrane, the modulation of root development, cold tolerance, and the resistance to *Botrytis cinerea*[86–89]. Studies on a distinct PM-localized CYBDOM from soybean expressed in *Xenopus* oocytes revealed a trans-plasma membrane electron flow dependent on both internal ascorbate and external electron acceptors such as FeCN or ferric nitrilotriacetate[63].

The current work shows that CRR is a CYBDOM protein with features consistent with an ascorbate-dependent $Fe^{3+}$ reductase. Structural analysis of the CRR CYB561 domain reveals a cytoplasmic basic pocket that could bind ascorbate as an electron donor. In addition, the CRR model shows not only the conserved histidines involved in heme biding in the CYB561 domain, but also the highly conserved histidine and methionine residues required in heme *b* binding in the DOMON domain of several bacterial and fungal proteins, and the plant AIR12[46,48]. It is likely that the addition of an extracellular DOMON domain to a CYB561 protein would enable it to transfer electrons to apoplastic acceptors, such as $Fe^{3+}$-chelates, located at a greater distance from the plasma membrane than would be possible with a classical transmembrane CYB561 protein. Alternatively, the DOMON domain could enable specific interactions with other substrates that will be reduced. Thus, although CRR can reduce various $Fe^{3+}$-chelates, it remains possible that other substrates could also be reduced which could ultimately impact the apoplastic ROS balance.

The *crr* mutant shares several phenotypes with the *pdr2*, *als3*, and *star1* mutants, including an enhanced reduction of primary root growth under −Pi, an increased Fe deposition in the apoplast of the SCN and dividing zone of the root meristem, and a reduction in meristem cell division[32,36,40]. In contrast to *pdr2*, *als3*, and *star1*, *crr* does not affect cell growth in the elongation zone, likely because the accumulation of apoplastic Fe in the elongation zone of *crr* is below a critical threshold. Although CRR has ferric reductase activity, it is unlikely to be involved in directly generating the local $Fe^{2+}$ required for LPR1 activity, since the *crr* mutant has opposite phenotypes compared to the *lpr1* mutant as far as root tip Fe accumulation and primary root growth under −Pi condition. We thus propose the following model for the implication of CRR on root growth and Fe homeostasis (Fig. 6e, left). Under P-limiting conditions, the expression of the malate transporter ALMT1 in the roots is induced as the result of STOP1 activation, leading to malate export to the root apoplast[35,38]. This, together with a de novo pool of $Fe^{3+}$ generated by the activity of the ferroxidase LPR1, promotes the accumulation of high levels of $Fe^{3+}$-malate in the root apoplast. $Fe^{3+}$-malate reduces division in the root apical meristem through potentially synergic pathways, including the generation of ROS, the activation of the CLE14/CLV2–PEPR2 pathway, and the interference with cell-to-cell communication by callose deposition[37,40]. Apoplastic $Fe^{3+}$-malate also reduces cell growth in the root elongation zone, potentially via the activation of peroxidases, cytochrome p450s, and other redox-active enzymes, leading to the modification of the cell wall structure[35,38]. Blue light is an important component of this process because it activates HY5 in the shoots, promoting its migration to the roots,

where it regulates the expression of *LPR1*[41]. In addition, blue light triggers photo-Fenton reactions, further inhibiting primary root growth upon their exposure to light[42]. In our model, the ascorbate-dependent ferric reductase activity of CRR antagonizes the activity of LPR1, modulating the ratio between the ferric and ferrous iron pools ($Fe^{3+}/Fe^{2+}$) in the root apoplast. In the *crr* mutant, the ratio of $Fe^{3+}/Fe^{2+}$ would be shifted toward $Fe^{3+}$, increasing the accumulation of $Fe^{3+}$-malate complexes in the apoplast and thereby inhibiting root growth. On the other hand, when *CRR* is expressed in the meristem, the pool of apoplastic $Fe^{3+}$-malate would decrease via ferric reduction. If the $Fe^{2+}$ generated by CRR is coupled with intracellular transport, it would then be unavailable for re-oxidation in the apoplast by LPR1 and thus reduce the inhibition of primary root growth under −Pi. CRR protein levels are partially dependent on the ER chaperones CNXs and PDR2 for proper protein expression, folding, and/or localization.

The ferric reductase activity of CRR, and of its homolog HYP1, would also explain the effect of *CRR* expression on a plant's tolerance or sensitivity to high external Fe (Fig. 6e, right). In the *crr* and *crr hyp1* mutants, the reduction of ferric reductase activity in the presence of high external Fe may restrict the transport of $Fe^{2+}$ into cells, which would in turn reduce the long-distance Fe transport from root to shoot, preventing its intracellular accumulation to cytotoxic levels in sensitive tissues. Although phenotypes associated with ectopic overexpression should be interpreted with care, the increased ferric reductase activity associated with the overexpression of CRR is likely the cause of the greater $Fe^{2+}$ transport into the roots and transfer to shoots observed in the CRR OE line, leading to cytotoxic levels of Fe and the production of ROS, negatively impacting plant fitness. Further studies will be required to reveal the expression patterns of *CRR* and *HYP1* in shoots and germinating seeds to understand their role in Fe homeostasis and their contribution to germination and seedling establishment under high external Fe.

In most organisms, $Fe^{3+}$ reduction is coupled to $Fe^{2+}$ import either directly via $Fe^{2+}$ transporters, such as IRT1 in plants, which form a complex with FRO2 and the proton pump AHA2 at the PM[72], or via a coupled ferroxidase/permease transport system, such as the Fet3−Ftr1 high-affinity Fe uptake system of fungi[90,91]. The current work raises the interesting question of how the $Fe^{2+}$ generated by CRR could be coupled to Fe transport. Since *IRT1* expression is strongly decreased under Pi-deficient conditions, it is unlikely to be involved in CRR-mediated intracellular $Fe^{2+}$ transport in the root meristem[13,15–17]. However, the potential role of IRT1 in the transport of $Fe^{2+}$ generated by CRR should be experimentally examined.

Considering the numerous interactions between Fe and Pi homeostasis, it is interesting to highlight that a genetic screen initially devised to study the response of roots to Pi deficiency has led to the identification of key genes likely to be primarily involved in Fe homeostasis, such as CRR and LPR1. Recent studies of the *LPR* gene family in both *A. thaliana* and rice have highlighted their essential contributions to Fe transport and Fe homeostasis in roots and shoots[92–95]. Future research on CRR, HYP1, and other members of the CYBDOM and CYB561 protein families should reveal their potential contribution to

Fe homeostasis and plant growth, which may be distinct depending on the tissue and environmental conditions.

## Methods

### Plant lines and growing conditions

All Arabidopsis (*Arabidopsis thaliana*) lines and mutants used in this work were in the Columbia (Col-0) background. The mutants and transgenic lines *pdr2-1*[34], *fro2*[71], *irt1*[96], 39OE[58], and pCYCB1::GUS[97] were previously described. The *hyp1* mutant (At5g35735) was provided by Ricardo Fabiano Giehl (IPK, Gatersleben, Germany). The T-DNA insertion lines *crr-1* (*Salk_202530*), *crr-2* (*Salkseq_039879.2*), *crr;h1* (*Salk_016284, Salk_013527c*), *almt1* (*Salk_009629*), and *lpr1* (*Salk_016297*) were obtained from the European Arabidopsis Stock Center (Nottingham, UK) and genotyped using the primers listed in Supplementary Data 8. All the other genetic constructs used in this work were generated using Infusion and Gateway technology using the primers described in Supplementary Data 8, with the pENTR2b (Thermo Fisher Scientific, Waltham, MA, USA) and pFAST vector series[98] as entry and destination vectors, respectively. Transgenic lines were generated via *Agrobacterium tumefaciens*-mediated transformations[99].

The seeds were germinated on vertical square plates containing one-sixth-strength Murashige and Skoog (MS) salts without Pi (Duchefa Biochemie, Haarlem, the Netherlands), 1% (w/v) sucrose, 0.8% (w/v) agar (Plant Agar from Duchefa Haarlem, the Netherlands), 0.5 g l$^{-1}$ of 2-(N-morpholino)ethanesulfonic acid, and buffered to pH 5.8 with KOH. The residual amount of phosphate present in this media was 20 µM. For the +Pi plates, the medium was supplemented with potassium phosphate buffer to obtain a final Pi concentration of 1 mM. The +Pi and −Pi media contained a final concentration of iron of 15 µM, approximately. For the Fe tolerance assays on plates, the +Pi or −Pi medium was supplemented with FeNaEDTA to the final concentration specified. To assess Fe tolerance in soil, the seeds were sown in soil or clay pellets and watered with tap water or 1/6-strength MS, respectively. The pH of soil and pellets was 5.2 and 6.2, respectively. Fe was supplemented as FeNaEDTA to the concentration indicated in the text. The plant phenotypes were analyzed after 15 days.

### Proteomic analysis

Col-0 and *cnx1 cnx2* mutant plants were grown in +Pi and −Pi media for seven days before their roots were harvested for a proteomic analysis. The tissues were homogenized in SDS-containing FASP buffer (4% SDS, 0.1 M DTT, 100 mM Tris pH 7.5), heated at 70 °C for 5 min, and allowed to cool at room temperature (RT). The cooled proteins were centrifuged for 20 min at 4000 rpm, after which the supernatant was recovered and frozen at −80 °C. The concentration of each extract was determined by gel electrophoresis, Coomassie blue staining, and total lane densitometry in comparison with a pre-quantitated total cell extract, and 50 µg of each tissue extract was subjected to further processing. Four replicates for each condition were prepared. The eight-plex iTRAQ approach[100], based on isobaric labeling and the measurement of reporter fragment ion intensities, was used for the relative quantitation. The extracts were proteolytically digested and iTRAQ-labeled according to the iFASP procedure[101]. More than 98.5% of the sequences of all samples had peptide spectrum matches determined by MASCOT (www.matrixscience.com) that corresponded to fully labeled iTRAQ peptides.

The labeled iTRAQ samples (8 × 50 µg) were pooled and desalted on SepPak C18 cartridges (Waters Corporation, Milford, MA, USA). Dried eluates were dissolved in 4 M urea with 0.1% ampholytes (pH 3–10; GE Healthcare, Chicago, IL, USA) and fractionated by off-gel focusing[102]. The 24 fractions obtained were desalted on a microC-18 96-well plate (Waters Corporation), dried, and resuspended in 0.1% formic acid with 3% (v/v) acetonitrile for LC-MS/MS analysis.

A data-dependent LC-MS/MS analysis was performed on the extracted peptide mixtures after digestion using a Fusion tri-hybrid orbitrap mass spectrometer (Thermo Fisher Scientific) interfaced through a nano-electrospray ion source to a RSLC 3000 HPLC system (Thermo Fisher Scientific). The peptides were separated on a reversed-phase Easy-spray PepMap nanocolumn (75 µM inner diameter × 50 cm, 2 µM particle size, 100 Å pore size; Thermo Fisher Scientific) with a 4–76% acetonitrile gradient in 0.1% formic acid (total time: 140 min). Full MS survey scans were performed at 120,000 resolution. In the data-dependent acquisition controlled by Xcalibur 3.0.63 software (Thermo Fisher Scientific), the most intense multiply charged precursor ions detected in the full MS survey scan were selected for "multi-notch" synchronous precursor selection[103] and MS3 fragmentation, which minimizes the co-isolation of near-isobaric precursors (maximum scan cycle: 3 s).

The data files were analyzed with MaxQuant 1.5.3.30[104] incorporating the Andromeda search engine[105]. The sequence database used for the search was the UNIPROT *Arabidopsis thaliana* proteome containing 27,242 sequences, in addition to a custom database containing most of the common environmental contaminants (keratins, trypsin, etc.). Both peptide and protein identifications were filtered at a 1% FDR relative to hits against a decoy database built by reversing the protein sequences. The raw reporter ion intensities generated by MaxQuant were used in all following steps leading to the quantification. Further data analysis was performed using the R statistical programming language version 2.15.2 (R Core Team, 2012) and the Perseus software[104].

### Root length quantification

The plants were grown as described above and the plates were imaged with a flatbed scanner (Epson Perfection V700 photo; Epson, Suwa, Japan). The root length was measured using Fiji (http://fiji.sc/Fiji) and the plugin Simple neurite tracer.

### Determination of meristem size and cell length

The roots were cleared with Clearsee and stained with Calcofluor White[106], and observed using confocal microscopy. The average cell length of the cortical cells in the differentiation zone was calculated. The root apical meristem size was determined as the distance from the QC to the first elongating cell.

### Confocal microscopy

Confocal microscopy was performed using a Zeiss LSM 880 Airyscan (Carl Zeiss). Calcofluor White was excited at 405 nm and detected at 425–475 nm. GFP was excited with a 488 nm laser and the emitted light was collected at 493–538 nm. The analysis of confocal and bright field microscopy images was done with the ZEN 2 (blue edition) software from Zeiss and Fiji 2.9.0 (https://imagej.net/software/fiji/).

### Measurement of Pi content

Pi was quantified using the ascorbate–molybdate assay[107]. Around 20 mg of shoot or root tissue was placed in water and subjected to at least three freeze-thaw cycles to release the Pi. The Pi content was quantified using a molybdate assay with a standard curve. The Pi concentration was normalized per gram of tissue.

### Beta-glucuronidase (GUS) assay

Col-0 and *crr-1* expressing the *pCYCB1::GUS* reporter cassette grown in +Pi and −Pi conditions were incubated in the GUS staining buffer (50 mM sodium phosphate, 2 mM potassium ferrocyanide, 2 mM potassium ferricyanide, 0.2% Triton X-100, and 2 mM of the GUS substrate 5-bromo-4-chloro-3-indolyl-b-D-glucuronic acid)[108]. The roots were imaged using a bright field stereomicroscope (Axio Zoom.V16; Carl Zeiss, Oberkochen, Germany).

## Measurement of ferric-chelate reductase activity in roots

Roots from seven-day-old seedlings grown on 1/6-strength MS plus Pi were used to measure Fe(III) chelate reductase activity. The root tissue was washed with a 100 mM Ca(NO$_3$)$_2$ solution and incubated for 1 h in the dark with 1 ml of reaction solution. For the reduction of Fe(III) NaEDTA and Fe(III)-Malate, 100 μM of either Fe(III)NaEDTA or Fe(III)-Malate and 500-μM ferrozine solution was added to each sample, and the production of purple-colored Fe(II)−ferrozine complexes was quantified by measuring the absorbance of the solution at 562 nm[58]. For the ferricyanide (FeCN) reduction, the reaction solution was 500 μM FeCN, and the reduction of ferricyanide was recorded by a spectrophotometer at 420 nm[109].

## Measurement of ferric-chelate reductase activity in yeast

*Saccharomyces cerevisiae* strain S288C Wt or *Δfre1Δfre2* cells[110] were used for the ferric-chelate reductase activity assay. For this purpose, the cells were transformed with the empty centromeric p415 vector[111] as a negative control, or the same vector carrying the *CRR* or yeast *FRE1* coding sequence under the control of the constitutive *GPD* promoter. For the assay, the cells were grown on solid SD/-LEU media for two days at 28 °C and then overnight at 22 °C in 5 ml liquid media with or without an ascorbate precursor (89 mg/100 ml L-galacto-γ-lactone) until an OD600 = 0.6 approximately[61,62]. The cells were washed twice with H$_2$O, and 1 ml of the resuspension was placed in a 2 ml tube for the analysis. The cells were collected by centrifugation and resuspended in 650 μl of assay buffer consisting of MES 0.05 M (pH 6.5) and 5% glucose supplemented with 500 μM ferrozine and 500 μM of either Fe(III)-EDTA, Fe(III)-malate or FeCN (for FeCN reduction ferrozine was omitted from the solution). After a 2-h incubation, an aliquot was collected to record the optical density at 600 nm and the rest was centrifuged to pellet the cells. The optical density of the supernatant was recorded at 562 nm or 420 nm for the Fe(III)-EDTA/Fe(III)-malate or FeCN reduction, respectively.

## Histochemical detection of Fe in seedlings

Fe accumulation in seedlings was assayed using Perls staining as indicated[112,113]. For Perls staining, the seedlings were incubated in 4 ml of 2% (v/v) HCl and 2% (w/v) potassium ferrocyanide for 30 min. The samples were washed with H$_2$O and incubated for 45 min with 4 ml of 10 mM NaN$_3$ and 0.3% H$_2$O$_2$ in methanol. After this step, the samples were washed with 100 mM Na-phosphate buffer (pH 7.4) and were either analyzed or subjected to a DAB intensification step. For the latter, the seedlings were incubated for 30 min in the same buffer containing 0.025% (w/v) DAB and 0.005% (v/v) H$_2$O$_2$ and washed two times with H$_2$O. Finally, the samples were cleared with chloral hydrate (1 g ml$^{-1}$; 15% glycerol) and analyzed with an optical microscope.

## $^{55}$Fe transport in Arabidopsis tissues

Plants were grown for 7 days in plates containing 1/6-strength MS supplemented with 0.8% (w/v) of agar and transferred to plates supplemented with 600 μM FeNaEDTA and 1.67 μCi ml$^{-1}$ of ferric $^{55}$Fe for four days. To avoid the contact between shoots and the media, a glass cover slide was placed between the shoot and the agar surface. After this, the root and shoot tissues were harvested separately into glass vials and washed for 10 min in 5 ml of 1/6-strength MS supplemented with 600 μM Fe(III)-EDTA with gentle shaking. This washing step ensures the removal of $^{55}$Fe from the media on the external surface of roots or shoots, while preserving the intracellular $^{55}$Fe as well as the $^{55}$Fe precipitated into cell walls. The samples were dried for two days at 60 °C before being digested with 0.2 ml perchloric acid at 90 °C for 8 h. The solution was cleared by adding 0.4 ml H$_2$O$_2$ and incubating overnight at 90 °C. For the $^{55}$Fe quantification, 4 ml of HionicFluor scintillating mixture (PerkinElmer, Waltham, MA, USA) was added to the samples, and the counts per minute were measured with a Tri-Carb 2800TR liquid scintillation counter (PerkinElmer)[78].

## ROS quantification

ROS quantification was performed according to a published protocol[114]. Briefly, ~40 mg of tissue was ground in liquid nitrogen and resuspended in 400 μl of phosphate buffer. The solution was cleared by centrifugation at 13,000 × *g* for 5 min at 4 °C, and an aliquot of 10 μl was mixed with 190 μl of carboxy-H2DCFDA in PBS (to a final concentration of 25 μM carboxy-H2DCFDA) and incubated for 30 min. The florescence emission was quantified as relative fluorescence units.

## ROS detection in root tips

Production of ROS was measured with the fluorogenic dye 20,70-dichlorodihydrofluorescein diacetate (carboxy-H2DCFDA), a cell-permeant compound[30,40]. In brief, seedlings were incubated in 100 mM Na-phosphate (pH 7.2) supplemented with 10 μM Carboxy-H2DCFDA (Invitrogen) for 15 min in darkness and subsequently imaged in 100 mM Na-phosphate (pH 7.2) using a Zeiss LSM 880 confocal laser-scanning microscope (excitation 488 nm, emission 535−565 nm).

## RNA extraction and qRT-PCR

Total RNA was extracted using the ReliaPrep RNA tissue miniprep system kit (Promega, Madison, WI, USA) and treated with RNase-free DNase according to the manufacturer's instructions. The first-strand cDNA was synthetized using M-MLV reverse transcriptase (Promega). The qRT-PCR was performed in a Quantstudio 3 thermocycler (Thermo Fisher Scientific) using the SYBR select master mix (Thermo Fisher Scientific) and the primers listed in Supplementary Data 8. *ACTIN2* (At3g18780) was used as a reference gene to normalize the gene expression data. Three biological replicates and two technical replicates were performed for each gene.

## RNA-seq analysis

Col-0, *crr-1* and CRR OE were grown in +Pi and −Pi conditions for seven days, as described above, and root tissues from 60 seedlings were pooled in each biological replicate, from which total RNA was extracted using Trizol (Thermo Fisher Scientific). RNA-seq was performed on three biological replicates per condition using Illumina (San Diego, CA, USA) technology, with more than 25 million reads obtained for each of them. Adapters were removed from the purity-filtered reads, which were trimmed for quality using Cutadapt v. 1.8[115]. Reads matching ribosomal RNA sequences were removed using fastq_screen v. 0.11.1 (https://www.bioinformatics.babraham.ac.uk/projects/fastq_screen/). The remaining reads were further filtered for low complexity with reaper v. 15-065[116] and aligned against the *Arabidopsis thaliana* TAIR10.39 genome using STAR v. 2.5.3a[117]. The number of read counts per gene locus was summarized with htseq-count v. 0.9.1[118] using the *Arabidopsis thaliana* TAIR10.39 gene annotations. The quality of the RNA-seq data alignment was assessed using RSeQC v. 2.3.7[119]. Genes with low counts were filtered out according to the rule of 1 count per million in at least one sample. The library sizes were scaled using TMM normalization and log-transformed into counts per million using the EdgeR package version 3.24.3[120]. Differential expression levels were computed with limma-trend approach[121] by fitting all samples into one linear model. A moderated t-test was used for each contrast and an F-test was used for the interaction. For each group of comparisons, the adjusted *p*-value is computed by the Benjamini−Hochberg method, controlling for the false discovery rate (FDR or adj.P.Val).

## Gene editing using CRISPR/Cas9

The vectors used to perform the CRISPR/Cas9-mediated *CRR* gene editing were previously described[122]. Briefly, sgRNAs were designed using CRISPR-P V2 (http://crispr.hzau.edu.cn/CRISPR2/) and cloned into the entry vectors pRUs. Three different sgRNAs were combined into the pSF278 intermediate vector, and finally recombined into the T-DNA destination vector PcUBi4-2::SpCas9 FastRed.

## Structural and comparative analysis of the CRR protein

The CRR protein model was downloaded from the AlphaFold database (entry Q9LSE7). The signal peptide (residues 1–25) was removed for comparison with the crystal structures of the DOMON (PDB ID: 1D7B) and CYB561 (PDB ID: 4O79) domains. Structural superimposition and structure-based sequence alignment were performed with Chimera v.1.14[123]. Electrostatic surface potentials were calculated with The PyMOL Molecular Graphics System, Version 2.0 Schrödinger, LLC. Secondary structure elements representation and residue conservation were generated with ESPript server (https://espript.ibcp.fr)[124].

## Statistics and reproducibility

Statistical details can be found in the figure legends and Source data file. Data are presented as mean ± SD. Exact $n$ and $p$-values are provided in the Source data file for both significant and nonsignificant results. Correction for multiple comparisons using statistical hypothesis testing was used when performing one- or two-way ANOVA. All the phenotypic analysis were characterized performing at least two independent experiments obtaining similar results. The RNA-seq and Proteomic analysis were performed using at least 3 biological replicates, each of them consisting of a pool of several seedlings. Statistical analysis and graphs were generated with GraphPad Prism 9.

## Accession numbers

The Arabidopsis accession numbers used in this work are the following: CRR, AT3G25290; CRR-H1, AT4G12980; HYP1, AT5G35735; PDR2, AT5G23630; LPR1, AT1G23010; ALMT1, AT1G08430; BHLH39, AT3G56980; FRO2, AT1G01580; IRT1, AT4G19690; CNX1, AT5G61790; CNX2, At5G07340.

## Reporting summary

Further information on research design is available in the Nature Portfolio Reporting Summary linked to this article.

## Data availability

Data associated with all figures and tables is provided in the Source data file as well as Supplementary data files. The RNA-seq datasets for *crr-1* and CRR OE have been deposited at the Gene Expression Omnibus with the accession number GSE235511 and GSE215755, respectively. The proteomic data from this work has been deposited at ProteomeXchange Consortium via the PRIDE partner repository with the dataset identifier PXD040981. Plant lines will be made available upon request to the corresponding author. Source data are provided with this paper.

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

## Acknowledgements

The authors acknowledge Manfredo Quadroni and Patrice Waridel of the Protein Analysis Facility and the team in the Genomic Technology Facility, University of Lausanne, Switzerland, for the proteomic and transcriptomic analyses, respectively. The support of the University of Lausanne Cellular Imaging Facility is also acknowledged. The authors are grateful to Ricardo Fabiano Giehl and Nicolaus von Wiren (IPK, Gatersleben, Germany) for providing seeds of the *hyp1* mutant, Thierry Desnos (CEA, Cadarache, France) and Petra Bauer (Heinrich Heine Universität, Düsseldorf, Germany) for sharing seeds of the *lpr1* and *almt* mutants, as well the 39 OE transgenic line, and Diane Ward (University of Utah, Salt Lake City, USA) for the yeast mutant Δ*fre1 fre2*. The authors are also grateful to Christian Dubos and Christian Hardtke for helpful discussions. This work was supported by a grant from the Swiss National Science Foundation (31003A-182462) to Y.P. Research in JS lab was financially supported by the University of Lausanne, the European Research Council (ERC) grant agreement no. 716358 and the Swiss National Science Foundation grant no. 310030_204526, while research in the CN lab was funded by the Deutsche Forschungsgemeinschaft grant 400681449/GRK2498.

## Author contributions

J.C. and Y.P. conceived the project. J.C. performed all experiments, with the exception of the root proteomics performed by J.M. and the analysis of ROS production in roots performed by C.N. P.J.S. performed the structural analysis and figures associated while J.S. supervised the structural analysis and associated text. J.C. and Y.P. wrote the manuscript, all authors provided feedback on it and approved the final manuscript. Y.P. agrees to serve as the author responsible for communication.

## Competing interests

The authors declare no competing interests.
