## [Peer Review File · Nature Communications]

A CYBDOM protein impacts iron homeostasis and primary root growth under phosphate deficiency in ArabidopsisREVIEWER COMMENTS

Reviewer #1 (Remarks to the Author):

The manuscript is well written and represents an important progress in the plant science community. I only a few suggestions and comments as shown below.

1. In line 77, "Strong evidence that" should be "Strong evidence indicate that".
2. In line 86, "FTI" should be "FIT".
3. In line 185, "did" should be "that in".
4. In line 151, "Among the seven candidates was a member of the CYBDOM family (AT3G25290)". The meaning of this sentence is not clear.
5. The authors use the $\Delta fre1 \Delta fre2$ strains to illustrate the ascorbate-dependent ferric reductase activity of CRR. It is a good evidence, but it might be better to add Fe³⁺-malate as an additional evidence if possible.

Reviewer #2 (Remarks to the Author):

Through a proteomic strategy aimed at identifying proteins differentially expressed under phosphate starvation, the authors identified CRR (a cytochrome b561 protein with an extracellular DOMON domain). They show in yeast that CRR displays an ascorbate-dependent ferric reductase activity. The primary root of the *crr ko* mutants is hypersensitive to Fe-dependent growth inhibition (under low-P condition), whereas in the CRR overexpressors it is hyposensitive; these responses are correlated with the level of accumulation of Fe in the root tip, especially around the stem cell niche. They show that CRR is a plasma membrane protein whose accumulation in the root tip needs PDR2.

The authors thus attribute a role to the CRR protein that is part of a family that has remained mysterious for many years. In itself, this result is noteworthy. Moreover, their analysis of epistasis between *crr* and *pdr2* mutants and between CRR OE and *pdr2* allows PDR2 to be placed upstream of LPR1 in this iron redox mechanism, where the single epistasis between *lpr1* and *pdr2* was undecidable until now. Thus, not only does this study reveal a new component in this pathway, but it also clarifies the relative position of some other steps.

Where this work opens up broader perspectives than the "simple" root growth in response to phosphate deficiency is the discovery of a role for CRR in iron homeostasis, in roots and leaves. The authors show that plants that overexpress CRR are hypersensitive to high iron concentration in the culture medium (in vitro and in soil) and this is correlated with the overaccumulation of iron in the shoot. Indeed, the CRR overexpressors transfer more Fe from roots to the shoot.

I really enjoyed reading this manuscript. The results are well presented, robust (as far as I can judge) and convincing.

I have only few questions, minor comments, and a suggestion.

I suggest the authors compare the predicted structure of CRR (as given by AlphaFold whose predictions are especially robust for membrane proteins) with the crystal structure of the human duodenal cytochrome b (Dcytb) (pdb number = 5zlg) or with that of Arabidopsis cytochrome b561 (pdb number = 4o79). These comparisons (superposition of 3D structures)

will further substantiate their hypothesis of CRR being a heme protein with a putative ascorbate-binding site on the cytoplasmic side.

AlphaFold is free, and a free version of the PyMOL software could be downloaded to make the superposition of 3D structures (using pdb files). This simple in silico experiment can be done quickly, even for someone who is not a structural biologist.

Questions and comments

Overexpression of CRR stimulates iron transfer from roots to cotyledons. What is the level of Fe accumulation in true leaves? Does a WT shoot grafted onto a CRR OE root overaccumulate more iron (than the WT)? And conversely, does a CRR OE shoot grafted onto a WT root altered in iron content? In other words, which organ is important for this transfer, the root, the leaves or both? (I expect at least the roots).

Line 642-645: "irt1 mutant behaves like the WT as far as the inhibition of primary root growth under $-Pi$ and apoplastic Fe deposition, it is unlikely to be involved in CRR-mediated Fe⁺² transport, at least in the root meristems ». I don't quite understand this conclusion. Fe-dependent root growth inhibition and iron transfer to the shoot are two different phenomena. The first phenomenon is a chemical reaction in the apoplast of cells localized at the root tip, while the second phenomenon requires iron to enter the symplasm. The first does not require IRT1 (as shown in this work and in Müller et al. 2015), but this does not necessarily eliminate a possible role for IRT1 in the second (although IRT1 is less expressed under low-Pi). Unless the authors have already tested this possibility with the double homozygous CRR OE, irt1, they should at least remain open with this hypothesis.

Fig. 4 A & B show that the non natural Fe-EDTA compound (used as a iron source in the plant growth media) seems to be a poor substrate for CRR OE (compared to FeCN). Did the authors have tested natural Fe complexes like Fe(III)-malate or Fe(III)-deoxymugineic acid?

Minor points:

Line 516: replace " μm " by " μM ".

Lines 610-614: Be careful when summarising these published data about the effect of blue light on root inhibition because the "HY5 model" and the "photo-Fenton model" are mutually exclusive.

Reviewer #3 (Remarks to the Author):

Clua et al. identified the CYBDOM protein CRR through proteomic analyses for candidate proteins with decreased abundance in a mutant altered in the response of root growth to Pi supply. They show that two allelic crr mutants have a short-root phenotype under $-Pi$ conditions, attributed to decreased meristem size. Ectopic overexpression of CRR (35S-CRR) results in meristem size increase and lengthened primary roots under $-Pi$, with root stem cell morphology and overall Perls/DAB-detectable Fe levels in these roots following broadly what is known from other mutants affecting this response. Genetic analyses suggest that lpr1 and PDR2 are epistatic to crr, and almt1 partially to a small degree, with CRR-GFP protein depending on PDR2. The authors report elevated extracellular ferric chelate

reductase activity in 35S-CRR plants, as well as a CRR-expressing yeast ferric reductase mutant, more with ferricyanide than with Fe-EDTA as an electron acceptor substrate, with an indication towards ascorbate as the electron donor. A transcriptomic comparison of Pi deficiency responses between roots of wild-type and 35S-CRR plants suggest a more iron-sufficient physiological state in the former. Yet, 35S-CRR1 plants are sensitive to excess Fe, and the data suggest that *crr* is more tolerant, correlating with Perls/DAB-accessible Fe and ROS levels in shoots and slightly higher levels of radiolabeled Fe accumulated in shoots over 4 d. The characterization of *crr* primary root growth phenotypes and genetic interactions is convincing and detailed, but the reviewer has some concerns about the work on CRR protein function and on the link between CRR1 and iron homeostasis in generating the root growth phenotypes. Introduction and discussion are very well written.

Major comments

1. The authors address the question of the connection between Arabidopsis CRR function and iron in Fe-dependent primary root growth inhibition under Pi deficiency largely in transgenic lines ectopically overexpressing CRR (35S-CRR) and under phosphate-sufficient iron-excess conditions. The reviewer wonders whether and in which way these data are relevant for the question addressed.

In addition, the reviewer feels it would be necessary to see also root or root tip ferric reductase activity (Fig. 4A), transcriptome data (Fig. 4D-E; unrealistic, admittedly) and Fe accumulation data (Fig. 6D) for *crr1* additionally, as well as primary root elongation under minus Pi for *crr hyp1*. Having data as shown in Fig. 6C for roots or root tips also, in particular including +Pi and -Pi cultivation conditions, would be insightful.

2. The -Pi phenotype of roots of 35S-CRR suggests that root physiological Fe status is less Fe-sufficient than that of the wild-type under this condition. This suggests that under Pi deficiency, CRR OE plants might perceive less internal Fe than the wild type. By contrast, the hypersensitivity of CRR-OE to excess Fe supply suggest rather that 35S-CRR take up or perceive more internal Fe. So it is a little unclear how the excess Fe data relate to the -Pi data, and - given that CRR is ectopically overexpressed - the line may provide information rather on putative protein function than on the physiological role of CRR. The *crr* mutant phenotypes under +Pi excess Fe are more interesting in this respect, but indicate inversely to the transcriptome data of 35S-CRR1 and also inversely to root tip Fe accumulation data (see also Fig. 6A). This should be discussed. Note that it is known that Fe deficiency responses of roots are systemically governed by the physiological iron status of the shoot.

3. Each time the authors used Fe either in growth media or as a substrate in an experiment (e.g. line 678, Fig. 4A and B, Fig. 5, S15, line 462, line 465, lines 474/475, line 496), they must specify in which chemical form (+II or +III) Fe was applied here ("Fe-EDTA" or "FeNaEDTA" omits this information, or - strictly speaking - "FeNaEDTA" means Fe(III)). In an oxic atmosphere, even if Fe was added as Fe(II), it will be rapidly oxidized to Fe(III), but Fe(III) is far less bioavailable and moves differently into cells and within a plant. So what did the authors do to keep Fe²⁺ (wherever they used it) in +2 form (e.g. flooding on soil, ascorbate addition to plate media or substrate solutions)? Fe precipitates as Pi salt but also as hydroxides or oxides - so medium and soil pH are also essential pieces of methodological information here. Not controlling the oxidation state of Fe in experiments - as may have occurred in this manuscript - complicates the interpretation of the results.

4. Some very important data are not shown in a replicated and quantitative/statistically documented manner, e.g. Fig. S8C (-Pi) (and here only a single transgenic line). In Fig 3D, for example, a reviewer would expect support from multiple frames taken upon identical cultivation, processing and imaging settings and a plot of quantitative differences with

statistics, or alternatively (in this case) an immunoblot. Similar for Fig. 1H and I, 2G, 6A, B, S3, S8B (-Pi), S9 (at least for CRR OE), S11, S15, S16).

5. Lines 515-520, Line 799 to 807 (55Fe uptake): According to the methods, the authors did not desorb apoplastically bound Fe – however, this is state-of-the-art. Fe binds highly effectively to external cell walls. To distinguish between apoplastic binding and uptake into cells/the plant, desorption is necessary, but the methods do not describe such as step. Moreover, if plants were grown on vertically oriented plates and their leaves touched the medium, this would be necessary for not only for roots but also for shoots in order to quantify their Fe uptake. This is also state-of-the-art. The methods text on the cultivation of plants for Fig. 6D does not contain this information of whether the leaves of the seedlings might have touched the medium containing radio-labeled Fe. Finally, the reviewer also feels that the Fe concentration used in this experiment was excessively (unrealistically) high (see methods). When provided at high levels, Fe can then enter cells non-specifically via a variety of membrane transporters. If deviation in such a way from the state-of-the-art, the authors must change the text to make this very explicit and rationalize it for the readers.

6. Line 278 to 288: In the eyes of the reviewer, this is a key experiment. Therefore, it is essential to explain and interpret here not only the results from the one promoter which enabled complementation but also the results from all promoters that did not. For example, the WOX5 promoter appears to govern expression in the same location as PR1, at least based on the images (but WOX5 should be active in QC and LPR1 in SCN according to S8A). According to Fig. 5A of Muller et al. 2015, the reviewer feels that the QC cells are the ones showing the highest LPR1 promoter activity. This needs clarification based on the actual images provided in this manuscript. Only pLPR1-CRR complements, but not pLPR1-CRR, which would suggest that an LPR1 expression domain outside the QC/SCN is decisive for complementation? So which one? In this context, it is critical in Fig. S8 to additionally show data exactly as in S8B, but for –Pi conditions. The reviewer feels that to be able to take advantage of this elegant approach, the SHR and maybe other promoters would have been very useful additionally because of the large LPR1 expression domain, which even expands under –Pi (Muller et al., 2015) and thus complicates the interpretation of this result. See also comment no. 4.

7. In the ferric reductase assays in Fig. 4A and 4B, the reader should be informed that the concentration of the ferricyanide substrate used was 5x higher than the concentration of the ferric EDTA substrate. The legends or results text, and also the interpretation of these data should take this into account.

8. The text emphasizes as motivation of the study (e.g. line 26, line 118-119, line 533) that under low Pi, the central role of LPR1/2 as ferroxidases in primary root growth attenuation implies the action of a ferric reductase to provide Fe²⁺ in the root meristem. If CRR were to fill this gap, one would then predict that the *crr1* mutant exhibits elevated root elongation under – Pi compared to the wild type, like *lpr1*. But this is not the case. The authors should rephrase this or explain in more detail their expectations on the role of CRR in this. Later, in the discussion (lines 614-618), the authors propose that CRR antagonizes the activity of LPR1 to modulate the Fe³⁺/Fe²⁺ ratio, whereby an elevated ratio would lead to increased levels of Fe³⁺-malate complexes and root growth inhibition. The reason how increased levels of Fe³⁺-malate would enhance ROS cycling and root growth inhibition should be clarified. Could the removal of Fe(III) from the root apoplast through reduction to Fe²⁺ and subsequent root-to-shoot transport, as proposed in the model, help, given that new Fe(III) in the soil solution would continuously have access to this site. In the cultivation system used

here plenty of Fe is present in the medium. Next, evidently, in $-Pi$ conditions, LPR1 alone has sufficient Fe^{2+} substrate available to do its job, even without CRR (or without HYP1) and possibly without both of them (?).

9. Statement lines 492 to 494. The observations described/summarized here seem to contradict the present model of how primary root growth stalls under Pi deficiency. As the authors correctly state here, staining intensity was near saturation in Fig. S16 in both $+Pi$ and $+Pi$ 350 μM Fe. Under $-Pi$, staining was even stronger (see Fig. 1I). The reviewer thus feels it is critical to provide data as shown in Fig. 6A and for the same genotypes, but grown in $-Pi$ conditions.

10. Line 503 to 505: See my comment on replication, quantification and statistics. It would be very important to have data for iron uptake of roots and shoots also for *crr* and *crr hyp1* in addition to the wild type and the overexpressor. The differences observed in 6D could be a consequence merely of ectopic overexpression and thus have limited potential for informing us on the physiological role of *crr*. The statement in lines 511-513 on Fe accumulation in roots and shoots of *crr* and *crr hyp1* mutants is not sufficiently supported by data at present.

Comments on both manuscripts jointly:

11. The reviewer wonders whether by supplying ferricyanide (Fig. 4A and B) as a substrate to CRR proteins, a previously reported general action of ferricyanide in protein-bound heme oxidation comes to bear, and whether the observed ferrereductase activities in this manuscript are thus an observation that does not correspond to the biological function of CRR. The reviewer wonders whether while CRR can donate an electron to Fe^{3+} under appropriately designed/alterd conditions, this is in fact a negligible process in planta under the usual range of physiological conditions including phosphate deficiency. The authors of both manuscripts should contemplate (and possibly provide reasons against) the following model: HYP1 or CRR might donate an electron to directly quench extracellular ROS and thus Fenton chemistry in the apoplast of the stem cell niche. There needs to be a fine balance though, because in principle and under "permissive" conditions, electron donation to apoplastic molecules via CRR or HYP1 (e.g. in overexpressing shoots) can alternatively enhance ROS production and thus promote Fenton chemistry. Next, evidently, in $-Pi$ conditions, LPR1/2 alone has sufficient Fe^{2+} substrate available to do its job, even without CRR (or without HYP1), and possibly without both of them (?), and thus LPR1/2 alone is able to account for the Fe-dependence of primary root growth inhibition phenotypes; the need for something additional (CRR/HYP1) to generate Fe^{2+} (in this case to moderate root growth inhibition) is not evident. Quantitative depletion of Fe(III) in the root apoplast through reduction to Fe^{2+} seems like a goal that is hard to achieve in a location that is permeable to the soil solution containing plenty of Fe(III).

12. How do the authors explain the root growth difference in *crr1 almt1* (barely longer/meristem size not larger than *crr*, only about 50% root length of *almt1*) compared to *hyp1 almt1* (like *almt1*) in the accompanying manuscript under $-Pi$?

13. The authors should provide results from the root growth assay conducted under phosphate-sufficient and -deficient conditions also for the *crr hyp1* double mutant, in addition to the single mutants and the wild type. Including this in both manuscripts will contribute to the comparability of both manuscripts because the media and cultivation systems were different.

14. The reviewer is struggling with the impression that plenty of Fe (see Fig. 2G, likely mostly Fe(III)) seems to be present near QC/SCN in +P conditions, but no stalling of primary root elongation occurs. While the presence of Fe is clearly necessary for this under –Pi conditions, the reviewer wonders whether much of what is observed several days later is merely a secondary and broader consequence of highly localized events that occurred within a much shorter period of time after the initiation of Pi deficiency.

15. The authors should consider in their models that the CYBDOM proteins may act somewhat independently in different parts of the plant – so in the opinion of the reviewer, if they generate an extracellular Fe²⁺ pool that is then taken up to contribute to shoot Fe levels, this Fe may not originate from the Fe²⁺ pool localized where stalling of root growth occurs under –Pi conditions. HYP1 appears to be present in a variety of cell types and even under +Pi conditions in a subset of these. This complicates pinning down the location of where the protein acts to contribute to foster primary root elongation under –Pi.

Minor comments:

16. Fig. S3A-B: transient localization of chimeric GFP-fusion proteins in a heterologous system is generally not as well accepted. The state-of-the art is to use the native promoter in stable transformant lines.

17. Methods line 675: Please specify the type of agar used. The authors are asked to provide the information of how much phosphate and iron (μM in the final medium) the agar and the sugar components added to their media.

18. The model (Fig. 6E) should clarify position(s) in the root. In the stem cell niche or in the root apical meristem, for example, the stele is not yet functional in root-so-shoot translocation yet. Could the authors also incorporate the action of CLE14/CLV2-PPR2 in the model or state a reason why this is omitted?

19. Fig. S10: reviewer is surprised that there are no effects of pH and H₂O₂ levels on –Pi response of the wild type (Col-0). How is this compatible with the model?

20. Lines 393-394: “show that CRR is” \diamond “show that CRR can act as”

21. Lines 423-424/and associated aspects: hard to grasp what interaction should mean; phrase by appropriate combinations of criteria for significant differences between genotypes AND between treatments?

22. Line 434: Please explain briefly the function of FRO8 here. Line 436: FRO4 not Fe-related (according to current knowledge)

23. Line 396 says that transcriptomics were done on roots, but Figure legend 4D says transcriptomics were done on plants. Please clarify.

24. FeCN: use or define at least once correct chemical formula. $[\text{Fe(III)(CN)}_6]^{3-}$?

25. Fe⁺²  Fe²⁺

26. Line 31: “CRR” should be in italics

27. Line 86: not FTI but FIT instead

28. Lines 121/122: please explain briefly in which direction calnexins influence Fe-dependent inhibition of primary root growth under Pi deficiency
29. Methods: add references for Perls/DAB
30. Line 235: 20% smaller meristem size (where is this shown?) Fig1DE?
31. Line 275: Abbreviation “SNC” must be defined; reviewer believes that this is probably a typo (should be “SNC”)
32. Line 549: adjust parentheses
33. Some of the information on numbers of replicates (“>=”) is not compatible with this journal’s instructions, to my knowledge.
34. Indicate what was normalized to in the Legend of Fig. S7A.
35. Mark stem cell niche cluster more explicitly in Fig. S7C
36. Figure S8A: first and third rows, third column “endodermis”.
37. Fig. S11 title: a “to” is missing.
38. Fig. S14D: add statistics?
39. Fig. S15D: y axis title “weight”

Reviewer #4 (Remarks to the Author):

growth under Pi deficiency. They generated a *cnx1/cnx2* double mutant, and they did a proteomic analysis to compare the proteomic changes between Col-0 and the mutant under Pi deficiency. After the identification of CRR, they further generated a *crr* mutant and the *crr* mutant displayed an enhanced reduction of primary root growth, which is linked with increased accumulation of Fe in the root meristem. Additionally, overexpression of CRR showed a minus Pi insensitive phenotype and regulated the genes in Fe homeostasis. I have some comments regarding this manuscript.

1. all the MS data and MaxQuant search results are needed to deposit into a public repository.
2. According to the proteomic analysis, CRR protein level in Col-0 roots did not display a significant change in response to Pi deficiency. If CRR is an important enzyme that regulates root growth under Pi deficiency, why the protein level did not increase under Pi deficiency? In Figure S4B, the mRNA level of CRR increased in Col-0 – Pi condition compared to +Pi condition. How do the authors explain this inconsistent result of mRNA and protein level?
3. Inconsistent result in Col-0 plus and minus Pi between transcriptomic and proteomic data. FRO4 showed increased in -Pi at transcriptional level, but FRO4 displayed a reduced protein level in -Pi compared to +pi in WT. How do the authors justify this result?

We appreciate the comments and suggestions from all reviewers since they contributed to improving the quality of our work. Below you will find a point-by-point response to all the comments in red font. In a similar way, all the changes made in the original manuscript can be found in red in the file with the suffix “highlight”.

REVIEWER COMMENTS

Reviewer #1 (Remarks to the Author):

The manuscript is well written and represents an important progress in the plant science community. I only a few suggestions and comments as shown below.

1. In line 77, “Strong evidence that” should be “Strong evidence indicate that”.

The sentence was amended as suggested (page 3, last paragraph)

2. In line 86, “FTI” should be “FIT”.

Typo was corrected (page 4, first paragraph).

3. In line 185, “did” should be “that in”.

Sentence was corrected as suggested (page 8, third paragraph).

4. In line 151, “Among the seven candidates was a member of the CYBDOM family (AT3G25290)”. The meaning of this sentence is not clear.

The sentence was changed to “Among the seven proteins differentially expressed in both conditions, there was a member of the CYBDOM family (AT3G25290)” (page 6, second paragraph).

5. The authors use the $\Delta fre1 \Delta fre2$ strains to illustrate the ascorbate-dependent ferric reductase activity of CRR. It is a good evidence, but it might be better to add Fe³⁺-malate as an additional evidence if possible.

As suggested by the reviewer, we performed new ferrireductase assays in roots and in yeast incorporating Fe-Malate as electron acceptor. Data was added to Figure 4A for plants, as well as 4B and Figure S21 for yeast. In plants, the results show that *CRR* over-expressing roots significantly reduce higher amounts of Fe(III)-Malate, Fe(III)-EDTA, and FeCN than the WT, with similar results for all three substrates (Figure 4A).

In yeast, *CRR* expression resulted in a 2-fold increase of Fe(III)-Malate reduction compared to the control $\Delta fre1 \Delta fre2$ double mutant (Figure S21), which although statistically significant, is clearly smaller than the reductase activity observed with FeCN. No reductase activity could be measured using Fe-EDTA in yeast. The apparent higher activity of CRR for FeCN (Figure 4B) when expressed in heterologous system is

consistent with previous data published by other groups (doi: 10.1104/pp.15.00642; 10.1016/j.febslet.2007.03.006). This observation might either reflect that CYBDOMs/CYB561 proteins reduce iron more efficiently when complexed with small chelators, or that specific physicochemical conditions of the extracellular matrix are required to reduce other forms of ferric iron. The new data is described in the Results section "CRR has ascorbate-dependent ferric reductase activity affecting root redox homeostasis" (pages 15-16).

Reviewer #2 (Remarks to the Author):

Through a proteomic strategy aimed at identifying proteins differentially expressed under phosphate starvation, the authors identified CRR (a cytochrome b561 protein with an extracellular DOMON domain). They show in yeast that CRR displays an ascorbate-dependent ferric reductase activity. The primary root of the *crr* ko mutants is hypersensitive to Fe-dependent growth inhibition (under low-P condition), whereas in the CRR overexpressors it is hyposensitive; these responses are correlated with the level of accumulation of Fe in the root tip, especially around the stem cell niche. They show that CRR is a plasma membrane protein whose accumulation in the root tip needs PDR2.

The authors thus attribute a role to the CRR protein that is part of a family that has remained mysterious for many years. In itself, this result is noteworthy. Moreover, their analysis of epistasis between *crr* and *pdr2* mutants and between CRR OE and *pdr2* allows PDR2 to be placed upstream of LPR1 in this iron redox mechanism, where the single epistasis between *lpr1* and *pdr2* was undecidable until now. Thus, not only does this study reveal a new component in this pathway, but it also clarifies the relative position of some other steps.

Where this work opens up broader perspectives than the "simple" root growth in response to phosphate deficiency is the discovery of a role for CRR in iron homeostasis, in roots and leaves. The authors show that plants that overexpress CRR are hypersensitive to high iron concentration in the culture medium (in vitro and in soil) and this is correlated with the overaccumulation of iron in the shoot. Indeed, the CRR overexpressors transfer more Fe from roots to the shoot.

I really enjoyed reading this manuscript. The results are well presented, robust (as far as I can judge) and convincing.

I have only few questions, minor comments, and a suggestion.

I suggest the authors compare the predicted structure of CRR (as given by AlphaFold whose predictions are especially robust for membrane proteins) with the crystal structure of the human duodenal cytochrome b (Dcytb) (pdb number = 5zlg) or with that of Arabidopsis cytochrome b561 (pdb number = 4o79). These comparisons (superposition of 3D structures) will further substantiate their hypothesis of CRR being a heme protein with a putative ascorbate-binding site on the cytoplasmic side.

AlphaFold is free, and a free version of the PyMOL software could be downloaded to make the superposition of 3D structures (using pdb files). This simple in silico experiment can be done quickly, even for someone who is not a structural biologist.

As suggested by the reviewer, we first performed CRR structure prediction using AlphaFold and we then superimposed its CYB561 domain with the one from the crystalized Arabidopsis CYB-1 (PDB 4O79). In addition, we superimposed CRR DOMON domain with the crystal structure of the *Phanerodontia chrysosporium* cellobiose dehydrogenase DOMON domain (PDB 1D7B). The comparison of CRR with these structures as well as the analysis of a structure-based sequence alignment of the CRR model and the crystal structures of the DOMON domain (PDB ID 1D7B, 6JT6 and 4QI3) and CYB561 domain (PDB ID 4O79 and 5ZLG) present in various proteins, strongly suggest that CRR binds one b heme in the apoplasmic DOMON domain as well as two b hemes in the transmembrane CYB561 domain, and that a positively charged ascorbic acid binding pocket is facing the cytosolic side of the CYB561 domain. All together, these data support the role of CRR as a ferrireductase transferring electrons from cytosolic ascorbate to apoplasmic ferrous iron using three b heme.

The analysis is shown in two new supplementary figures (Supplemental Fig. S3 and S4), the description of the results was added (page 7, second paragraph), some elements added in the Discussion (page 23, last paragraph) and a new section in material and methods describe the procedures used for the analysis (page 33, penultimate paragraph).

Questions and comments

Overexpression of CRR stimulates iron transfer from roots to cotyledons. What is the level of Fe accumulation in true leaves?

To answer this question, we performed a Perls-DAB staining on 11 dpg Arabidopsis seedlings, a stage where true leaves are developed. For this, plants were grown either for 11 days in +Pi or +Pi with 200 μ M Fe. As shown in the new Figure S32, CRR OE line clearly showed an hyperaccumulation of iron in true leaves under iron stress. A detailed description of the results was added (page 20, last paragraph).

Does a WT shoot grafted onto a CRR OE root over-accumulate more iron (than the WT)? And conversely, does a CRR OE shoot grafted onto a WT root altered in iron content? In other words, which organ is important for this transfer, the root, the leaves or both? (I expect at least the roots).

To address this question, we performed homo- and hetero-grafts of Col-0 and CRR OE seedlings and grew them under a high iron stress in order to identify clear phenotypes associated with iron toxicity (new Supplemental Figure S33). The results show that the hypersensitivity to iron stress observed in CRR OE lines is a consequence of CRR over-expression in roots **and** shoots. This data suggests that iron reduction performed by CRR is not only important for its uptake from the media, but also for its allocation in shoots

and likely also cellular internalization. A paragraph was added explaining these results (Page 21, last paragraph).

Line 642-645: “*irt1* mutant behaves like the WT as far as the inhibition of primary root growth under $-P_i$ and apoplastic Fe deposition, it is unlikely to be involved in CRR-mediated Fe^{+2} transport, at least in the root meristems ». I don't quite understand this conclusion. Fe-dependent root growth inhibition and iron transfer to the shoot are two different phenomena. The first phenomenon is a chemical reaction in the apoplast of cells localized at the root tip, while the second phenomenon requires iron to enter the symplasm. The first does not require IRT1 (as shown in this work and in Müller et al. 2015), but this does not necessarily eliminate a possible role for IRT1 in the second (although IRT1 is less expressed under low- P_i). Unless the authors have already tested this possibility with the double homozygous CRR OE,*irt1*, they should at least remain open with this hypothesis.

The reasoning was that the decrease in apoplastic iron observed in CRR OE meristem could be a consequence of iron reduction coupled to its cellular internalization, which in turn would reduce the amount of iron in the apoplast. The fact that *irt1* mutant didn't show any root phenotype under P_i deficiency, either for primary root growth, and that its expression was not detected in the root apical meristem, strongly suggest that IRT1 is not involved in the primary root growth inhibition under P_i deficiency. However, we totally agree with the Reviewer that IRT1 could still be associated with CRR OE hyper-sensitivity to iron toxicity, which is independent from the effect in the root meristem. Thus, we modify the paragraph highlighted by the reviewer in order to clarify the idea and avoid ambiguities (page 25, penultimate paragraph).

Fig. 4 A & B show that the non natural Fe-EDTA compound (used as an iron source in the plant growth media) seems to be a poor substrate for CRR OE (compared to FeCN). Did the authors have tested natural Fe complexes like Fe(III)-malate or Fe(III)-deoxymugineic acid?

We have now compared the ferric reductase activity using FeCN, Fe-EDTA, and Fe-malate (data was added to Figure 4A). In plants, we have compared the root ferric reductase activity in Col, *crr-1*, *crr-1 hyp1*, CRR OE and the 39 OE line previously characterized to have high ferric reductase activity due to the strong activation of the *FRO2* (Figure 4A). Overexpression of *CRR* (line CRR OE) showed a clear increase in ferric reductase activity with all three tested substrates (Figure 4A). Although the *crr-1* single mutant does not show a significant difference in reductase activity with any of the tested substrate, the double mutant *crr-1 hyp1* showed a statistically significant reduction with both Fe-EDTA and Fe-malate. This is likely due to the limited expression domain of the *CRR* gene when considering whole roots, while the combination of both *CRR* and *HYP1* covers a larger domain.

We also tested Fe-Malate reduction in yeast (Figure S21), however, CRR over-expression was only translated in a small increase of Fe(III)-Malate reduction compared to the $\Delta fre1 \Delta fre2$ double mutant (albeit statistically significant). The higher activity for FeCN shown by CRR in heterologous system (Figure 4B) is

consistent with data reported by other groups (doi: 10.1104/pp.15.00642; 10.1016/j.febslet.2007.03.006). This observation might either reflect that CYBDOMs/CYB561 proteins reduce iron more efficiently when complexed with small chelators, or that specific physicochemical conditions of the extracellular space are required to reduce other forms of ferric iron. The new data is described in the Results section “CRR has ascorbate-dependent ferric reductase activity affecting root redox homeostasis” (pages 15-16).

Minor points:

Line 516: replace “ μm ” by “ μM ”.

Typo was corrected (page 21, second paragraph).

Lines 610-614: Be careful when summarising these published data about the effect of blue light on root inhibition because the “HY5 model” and the “photo-Fenton model” are mutually exclusive.

We agree with the reviewer that the authors of both papers stated their models in a mutually exclusive way. However, as we discuss in the manuscript, our interpretation of their results is that both models can co-exist, and their relative importance might depend on the experimental conditions. Blue light can be required to promote HY5 migration and LPR1 regulation, and additionally, can enhance Fenton reactions taking place in the root apical meristem. Similar conclusions have also been proposed by Neuman and colleagues (Naumann et al., 2022, *Current Biology* 32, 2189–2205).

Reviewer #3 (Remarks to the Author):

Clua et al. identified the CYBDOM protein CRR through proteomic analyses for candidate proteins with decreased abundance in a mutant altered in the response of root growth to Pi supply. They show that two allelic *crr* mutants have a short-root phenotype under $-\text{Pi}$ conditions, attributed to decreased meristem size. Ectopic overexpression of CRR (35S-CRR) results in meristem size increase and lengthened primary roots under $-\text{Pi}$, with root stem cell morphology and overall Perls/DAB-detectable Fe levels in these roots following broadly what is known from other mutants affecting this response. Genetic analyses suggest that *lpr1* and *PDR2* are epistatic to *crr*, and *almt1* partially to a small degree, with CRR-GFP protein depending on *PDR2*. The authors report elevated extracellular ferric chelate reductase activity in 35S-CRR plants, as well as a CRR-expressing yeast ferric reductase mutant, more with ferricyanide than with Fe-EDTA as an electron acceptor substrate, with an indication towards ascorbate as the electron donor. A transcriptomic comparison of Pi deficiency responses between roots of wild-type and 35S-CRR plants suggest a more iron-sufficient physiological state in the former. Yet, 35S-CRR1 plants are sensitive to excess Fe, and the data suggest that *crr* is more tolerant, correlating with Perls/DAB-accessible Fe and ROS levels in shoots

and slightly higher levels of radiolabeled Fe accumulated in shoots over 4 d. The characterization of *crr* primary root growth phenotypes and genetic interactions is convincing and detailed, but the reviewer has some concerns about the work on CRR protein function and on the link between CRR1 and iron homeostasis in generating the root growth phenotypes. Introduction and discussion are very well written.

Major comments

1. The authors address the question of the connection between Arabidopsis CRR function and iron in Fe-dependent primary root growth inhibition under Pi deficiency largely in transgenic lines ectopically overexpressing CRR (35S-CRR) and under phosphate-sufficient iron-excess conditions. The reviewer wonders whether and in which way these data are relevant for the question addressed.

We believe that our data unveiled the biological function of CRR and its role in the primary root growth inhibition under phosphate deficiency, which is correlated with changes in iron accumulation in the root apical meristem. In addition, we unveiled that CRR and HYP1 are involved in iron homeostasis as well as affecting iron toxicity tolerance. In the original manuscript, we described strong and clear phenotypes not only using CRR over-expressing lines, but also the *crr* and *crr hyp1* mutants (e.g. on the primary root growth to Pi deficiency as well as response to Fe toxicity). In this revised manuscript, we now include additional data using the *crr* and *crr hyp1* mutants, namely:

-Root transcriptome of *crr* mutant grown in plus and minus Pi.

-Root phenotype of *crr hyp1* double mutant.

-ROS production in the root apical meristem under control and Pi deficiency conditions for *crr* and *crr hyp1*.

-Root ferric reductase assay in *crr* and *crr hyp1* double mutant.

-Iron transport assays in *crr* and *crr hyp1* double mutant.

While we understand the limits of interpreting overexpression data by themselves, we believe the results obtained using the CRR OE lines are significant and important when taken together with the results obtained with the *crr* and *crr hyp1* mutants and as such consolidate the implication of CRR in Fe reduction and Fe homeostasis in addition to its role in the response of primary root growth under Pi deficiency.

In addition, the reviewer feels it would be necessary to see also root or root tip ferric reductase activity (Fig. 4A), transcriptome data (Fig. 4D-E; unrealistic, admittedly) and Fe accumulation data (Fig. 6D) for *crr1* additionally, as well as primary root elongation under minus Pi for *crr hyp1*.

-We now include in Fig. 4A ferrereductase activity data for the *crr* and *crr hyp1* mutants using FeEDTA, FeCN, and Fe-malate. Our data shows that *crr* roots don't show a statistically significant reduction in ferrereductase activity compared to WT, however, the *crr hyp1* double mutant showed a small but significant reduction in ferrereductase when Fe-Malate and Fe-EDTA were used as electron acceptors. The difference

between *crr* and *crr hyp1* ferrireductase activity is likely due to the limited expression domain of the *CRR* gene when considering whole roots while the combination of both *CRR* and *HYP1* cover a larger domain. Overexpression of *CRR* (line *CRR OE*) showed a clear increase in ferric reductase activity with all three tested substrates (Figure 4A).

-We performed a transcriptomic experiment using *CRR OE* line to assess the effect that *CRR* overexpression triggers in roots to infer *CRR* molecular function and the cause of root insensitivity to Pi deficiency displayed by this line. Considering that this was the aim of the experiment, we don't believe that data in Fig. 4D-E was unrealistic (or at least uninformative). For example, it allowed us to conclude that the roots of the *CRR OE* line were not failing to reduce primary root growth under Pi deficiency because of a failure to sense Pi deficiency in a systemic fashion or fail to activate the Phosphate Starvation Responsive (PSR) genes. We assume that the point that the Reviewer wants to raise is that the transcriptomic analysis we made will not reveal *CRR* endogenous biological role since the gene was over-expressed, and we agree with this. We were somewhat skeptical about doing an RNA-seq of roots using *crr* mutant since the gene is expressed only in a limited region of the root and we thought that the effect of its mutation in the transcriptome could be too diluted. However, based on the Reviewer's comments we decided to perform the experiment and we think that the results are informative.

The data is now available at the Gene Expression Omnibus with the Id: GSE235511 and the analysis is now shown in Figures 4D, 4F, 4H, 4J, S22-26, and tables S2-7. A description of the results and the interpretation is described (pages 16-18). Briefly, the comparison between Col-0 and *crr* mutant showed that *crr* roots activate the general PSR genes in a similar fashion to Col-0 or *CRR OE* under -Pi. The differentially expressed genes under Pi deficiency (compared to WT), showed a strong enrichment in categories related to 'redox' and iron, supporting the hypothesis that *crr* short root phenotype is a consequence of altered iron accumulation in the root tip with triggers ROS production.

- In Figure 6D we now include data of Fe accumulation and transport for *crr* and *crr hyp1*. In either case, we didn't find significant differences in ⁵⁵Fe accumulation in shoots, roots, or the fraction of iron accumulating to shoots. These data suggest that *crr hyp1* strong tolerance to iron stress might be a consequence of more subtle differences in iron compartmentation, an aspect which cannot be revealed in these experiments. These results are now discussed (page 21, second paragraph).

-Finally, as the reviewer suggests, we assessed *crr hyp1* root elongation in +Pi and -Pi in parallel with Col-0, *crr*, and *hyp1* genotypes. Data is displayed in Figure S8 and the results discussed (page 9, first paragraph). The results showed that *crr hyp1* double mutant phenotype is stronger than either single mutants, suggesting that *CRR* and *HYP1* regulate root growth inhibition under Pi deficiency in a partially redundant manner.

Having data as shown in Fig. 6C for roots or root tips also, in particular including +Pi and -Pi cultivation conditions, would be insightful.

As suggested by the reviewer, we analyzed ROS production in root tips of Col-0, *crr*, *crr hyp1*, and CRR OE lines under +Pi and -Pi. This was made using the ROS probe Carboxy-H₂DCFDA. This data is now shown in Figure S27. Results showed that, as expected, *crr* and *crr hyp1* produced a higher amount of ROS in the SCN and root apical meristem in accordance with their short root phenotype. On the other hand, CRR OE line showed ROS levels comparable to a +Pi condition, as expected for the absence of root growth inhibition of this line under -Pi. Results are discussed in pages 18, second paragraph).

2. The -Pi phenotype of roots of 35S-CRR suggests that root physiological Fe status is less Fe-sufficient than that of the wild-type under this condition. This suggests that under Pi deficiency, CRR OE plants might perceive less internal Fe than the wild type. By contrast, the hypersensitivity of CRR-OE to excess Fe supply suggest rather that 35S-CRR take up or perceive more internal Fe.

So it is a little unclear how the excess Fe data relate to the -Pi data, and – given that CRR is ectopically overexpressed – the line may provide information rather on putative protein function than on the physiological role of CRR. The *crr* mutant phenotypes under +Pi excess Fe are more interesting in this respect, but indicate inversely to the transcriptome data of 35S-CRR1 and also inversely to root tip Fe accumulation data (see also Fig. 6A). This should be discussed. Note that it is known that Fe deficiency responses of roots are systemically governed by the physiological iron status of the shoot.

Under Pi deficiency CRR OE line shows a reduced accumulation of Fe in the root apical meristem. Whether or not this is translated into a general less Fe-sufficient status was not addressed in this works and is rather speculative. We never observed any symptoms of Fe deficiency (e.g. chlorosis in leaves) in CRR OE grown in +Pi or -Pi conditions. Under iron excess stress, however, plenty of iron is available and reduced in the whole root of CRR OE line, which is translated into a higher rate of Fe transport to shoots.

Arabidopsis primary root growth inhibition under Pi deficiency has been associated with a specific apoplastic iron accumulation at the root tip, specifically at the SCN and division zone of the meristem. Most of the mutants affected in this developmental process have been shown to be affected in the intensity of iron accumulation in this region of the root. Hence, CRR OE and *crr* root phenotypes under -Pi can be explained by the observed iron localization pattern in the meristem. Besides a mild deregulation of a few iron homeostatic genes in CRR OE roots, we didn't observe any strong indication of iron deficiency (or the perception of low internal iron status) in this line under Pi deficiency.

Iron toxicity is a different abiotic stress. In these conditions, our data suggest that CRR OE roots accumulate less apoplastic iron and transfer more iron to the shoots than Col-0. Our interpretation of this data, in conjunction with the phenotypes of the *crr* and *crr hyp1* mutants on Fe accumulation in the root meristem, is that CRR is linked to Fe homeostasis.

The data obtained with the CRR OE line should not be over-interpreted, in particular should not be used to precisely assign a function for CRR in shoots. Again, the data of the CRR OE should be look at in the perspective of what we also learned using the *crr* and *crr hyp1* lines. We tried to make this clearer in the discussion of the paper (page 25, second paragraph).

3. Each time the authors used Fe either in growth media or as a substrate in an experiment (e.g. line 678, Fig. 4A and B, Fig. 5, S15, line 462, line 465, lines 474/475, line 496), they must specify in which chemical form (+II or +III) Fe was applied here (“Fe-EDTA” or “FeNaEDTA” omits this information, or - strictly speaking - “FeNaEDTA” means Fe(III)). In an oxic atmosphere, even if Fe was added as Fe(II), it will be rapidly oxidized to Fe(III), but Fe(III) is far less bioavailable and moves differently into cells and within a plant. So what did the authors do to keep Fe²⁺ (wherever they used it) in +2 form (e.g. flooding on soil, ascorbate addition to plate media or substrate solutions)? Fe precipitates as Pi salt but also as hydroxides or oxides – so medium and soil pH are also essential pieces of methodological information here. Not controlling the oxidation state of Fe in experiments – as may have occurred in this manuscript – complicates the interpretation of the results.

In all the experiments aimed to trigger iron stress in plants, it was used the ferric iron (III) form, either as Fe(III)NaEDTA, [Fe(III)(CN)₆]³⁻, or Fe(III)-Malate. This has now been specified throughout the manuscript. We agree with the author that the pH affects iron solubility and hence it is an important piece of methodological information. The pH of the agar medium was already mentioned (page 27) (pH was 5.8). In addition, we now included information about the pH of soil and clay pellets (5.2 and 6.2, respectively) (page 27).

It is true that iron oxidation state can change in an oxic atmosphere, as well as in the presence of light due to photoreduction. To our knowledge, there is no way to control these phenomena, which also occur in natural environments. We believe, however, that our data is not affected by these changes in iron oxidation state since the phenotypes of the different lines were always analyzed in the same media and in parallel. In addition, the iron stress related phenotypes were reproduced in agar media exposed to continuous light, as well as in soil and clay pellets under a 16/8 hs photoperiod. This strongly suggest that the experimental conditions do not significantly affect the phenotypes of the mutants and transgenic lines compared to Col-0.

4. Some very important data are not shown in a replicated and quantitative/statistically documented manner, e.g. Fig. S8C (-Pi) (and here only a single transgenic line). In Fig 3D, for example, a reviewer would expect support from multiple frames taken upon identical cultivation, processing and imaging settings and a plot of quantitative differences with statistics, or alternatively (in this case) an immunoblot. Similar for Fig. 1H and I, 2G, 6A, B, S3, S8B (-Pi), S9 (at least for CRR OE), S11, S15, S16).

-Former figure S8C has been replaced with S15. We now show in S15 quantitative data for 2 independent lines for each genotype.

-As suggested, in Fig 3D we now provide quantitative data of CRR-GFP expression levels as mean intensity values (new bottom panel Figure 3E) .

-Fig. 1H shows a GUS staining which is rather difficult to quantify, considering that one must consider not only the signal intensity but also the spatial distribution of the signal. However, the qualitative differences between Col-0 and *crr* are very clear. To show the reproducibility of the GUS data, we now provide a supplementary figure showing images from three additional independent sets of roots (Figure S10), confirming our conclusions.

-Fig. 1I, 2G, 6A, 6B, and S16 show Perls or Perls-DAB staining pictures. This technique has been extensively used by the scientific community to assess differences in iron accumulation in a qualitative way in numerous publications. Again here, one must not only focus on signal intensity but also on signal distribution, making its quantification difficult (even potentially mis-leading). Thus, we prefer to keep with a qualitative description of such data, as found in other all publication using this technique. To highlight the robustness of the technique and our results, we now show multiple pictures from independent sets of roots, shoots or cotyledons in supplemental Figures, using Perls with or without DAB intensification (see Supplemental Figure S11, S12, S17, S29, S30, S31)

-Fig S3 shows the subcellular localization of CRR-GFP. We now plotted the CRR-GFP, PI or subcellular markers intensity signal profile to show their co-localization (now Figure S5). In addition, we quantified the GFP signal for panel (C) to quantitatively assess the difference between the cytosolic and secreted GFP1-10.

-Fig. S8B shows CRR expression pattern under the control of different cell-type specific promoters. The images show that the expression patterns we obtained fit with what was expected. We think that the reviewer's comment was actually related to panel C, hence we now show the root length quantification of all the lines (now Figure S15).

-Fig. S9 (now Figure S15) shows the meristem length of Col-0, CRR OE lines, and *lpr1*. The quantification of this was already shown in Fig. 2C.

-We now show the root length quantification related to Fig. S11 (now Figure S19).

-Fig. S15, now Figure S28, panel B is the quantification of A, panel D is the quantification of C. We now added the quantification of panel E as % of germination and survival in panel F and G of Figure S28.

5. Lines 515-520, Line 799 to 807 (55Fe uptake): According to the methods, the authors did not desorb apoplastically bound Fe – however, this is state-of-the-art. Fe binds highly effectively to external cell walls. To distinguish between apoplastic binding and uptake into cells/the plant, desorption is necessary, but the

methods do not describe such as step. Moreover, if plants were grown on vertically oriented plates and their leaves touched the medium, this would be necessary for not only for roots but also for shoots in order to quantify their Fe uptake. This is also state-of-the-art. The methods text on the cultivation of plants for Fig. 6D does not contain this information of whether the leaves of the seedlings might have touched the medium containing radio-labeled Fe. Finally, the reviewer also feels that the Fe concentration used in this experiment was excessively (unrealistically) high (see methods). When provided at high levels, Fe can then enter cells non-specifically via a variety of membrane transporters. If deviation in such a way from the state-of-the-art, the authors must change the text to make this very explicit and rationalize it for the readers.

In all the $^{55}\text{Fe}^{+3}$ experiments reported in this work, the roots exposed to $^{55}\text{Fe}^{+3}$ were extensively washed with a solution containing non-radioactive Fe^{+3} before counting. This step should have allowed some level of exchange of apoplastically bound $^{55}\text{Fe}^{+3}$ in the roots with non-radioactive Fe^{+3} . Furthermore, in all $^{55}\text{Fe}^{+3}$ experiments, care was taken so that the shoot does not come into contact with the media by placing a glass coverslide below the leaves, thus ensuring that the ^{55}Fe detected in shoots came only from its transfer from roots. These elements have now been clarified in the methods (page 31). Finally, the amount of non-radioactive Fe used in these experiments was chosen because it corresponds to the amount of Fe where a clear phenotype is observed for the CRR OE lines under these particular growth conditions of 7 days on MS medium followed by 4 days of growth in a high (600 μM) Fe medium, as shown in Figure S28C, D. The aim of the experiment being to understand how this Fe-toxicity phenotype observed in the CRR OE occurred, we used the same concentration of Fe in the $^{55}\text{Fe}^{+3}$ uptake and transfer experiments.

6. Line 278 to 288: In the eyes of the reviewer, this is a key experiment. Therefore, it is essential to explain and interpret here not only the results from the one promoter which enabled complementation but also the results from all promoters that did not. For example, the WOX5 promoter appears to govern expression in the same location as PR1, at least based on the images (but WOX5 should be active in QC and LPR1 in SCN according to S8A). According to Fig. 5A of Muller et al. 2015, the reviewer feels that the QC cells are the ones showing the highest LPR1 promoter activity. This needs clarification based on the actual images provided in this manuscript. Only pLPR1-CRR complements, but not pLPR1-CRR, which would suggest that an LPR1 expression domain outside the QC/SCN is decisive for complementation? So which one? In this context, it is critical in Fig. S8 to additionally show data exactly as in S8B, but for $-\text{Pi}$ conditions. The reviewer feels that to be able to take advantage of this elegant approach, the SHR and maybe other promoters would have been very useful additionally because of the large LPR1 expression domain, which even expands under $-\text{Pi}$ (Muller et al., 2015) and thus complicates the interpretation of this result. See also comment no. 4.

In response to the comments of the reviewer we have added results obtained with the SHR promoter, as well as examine the expression pattern of the various promoter-GFP constructs under $-\text{Pi}$ conditions (Figure S14). For each promoter tested, there is no difference in the tissue-specific pattern observed

between +Pi and -Pi with the notable interesting exception of the WOX5 promoter. While WOX5 give a clear expression pattern in QC under +Pi condition, under -Pi condition, expression in QC appear lost or strongly decreased and expression in cortical cells of the meristematic zone appears.

From all the promoters tested, only LPR1:CRR-GFP complements the short primary root growth phenotype under -Pi condition. Under this condition, only LPR1 is expressed in the QC and neighboring initials. While it is true that LPR1 allow expression of CRR most strongly in the QC and initials, there is also weaker but clear expression in a broader meristematic zone that encompass the endodermis, cortical and even epidermal cells (Figure S14) Similar broader expression pattern extending beyond the SCN is also found in Muller et al. 2015. We can conclude from the use of the other promoters that expression of CRR only in the pericycle initials (ATL75 promoter), the meristematic vasculature (SHR, WOL), meristematic or elongating endoderm (SCR, MYB36), meristematic cortical cells (pCO2), or root cap (pLOVE) is not sufficient for complementation. Since WOX5 lost its QC-specific expression pattern under Pi deficiency, we cannot determine unambiguously if expression of CRR in the QC only would be sufficient for complementation or if expression in neighboring initial is also required. But clearly expression of CRR-GFP in the QC in *crr* plants grown on +Pi is not sufficient to prevent the short-root phenotype when *crr* mutant are grown in -Pi. So we would conclude that complementation cannot be achieved by expression in only QC and initials but that a broader expression in the meristematic zone is required. Such a conclusion would be supported by the Fe accumulation pattern observed in the *crr* mutant under -Pi condition (Figure 1i) . We now explain these points in more detail (page 11-12).

7. In the ferric reductase assays in Fig. 4A and 4B, the reader should be informed that the concentration of the ferricyanide substrate used was 5x higher than the concentration of the ferric EDTA substrate. The legends or results text, and also the interpretation of these data should take this into account.

In the root ferrereductase assay shown in Figure 4A we used different concentrations of iron since the quantification methods for FeCN and FeEDTA are different. Thus, we followed specific protocols for each substrate. Even though the information describing the concentrations was in material and methods, we now also included it in the legend of the figure. We would note that we draw no specific conclusions as far as the relative difference in ferric reductase activity detected in plants between FeCN and Fe-EDTA (make conclusions only between different genotypes).

The yeast ferrereductase assay experiments shown in Figure 4B and S21 were now redone using the same buffer and iron concentrations for FeCN, Fe-EDTA, and Fe-Malate, i.e. 500 μ M, based on comments from the other reviewers. This is now indicated in the Figure legends as well as Material and Methods (page 30). Data can be found in figure 4B and S21.

8. The text emphasizes as motivation of the study (e.g. line 26, line 118-119, line 533) that under low Pi, the central role of LPR1/2 as ferroxidases in primary root growth attenuation implies the action of a ferric

reductase to provide Fe²⁺ in the root meristem. If CRR were to fill this gap, one would then predict that the *crr1* mutant exhibits elevated root elongation under – Pi compared to the wild type, like *lpr1*. But this is not the case. The authors should rephrase this or explain in more detail their expectations on the role of CRR in this. Later, in the discussion (lines 614-618), the authors propose that CRR antagonizes the activity of LPR1 to modulate the Fe³⁺/Fe²⁺ ratio, whereby an elevated ratio would lead to increased levels of Fe³⁺-malate complexes and root growth inhibition. The reason how increased levels of Fe³⁺-malate would enhance ROS cycling and root growth inhibition should be clarified. Could the removal of Fe(III) from the root apoplast through reduction to Fe²⁺ and subsequent root-to-shoot transport, as proposed in the model, help, given that new Fe(III) in the soil solution would continuously have access to this site. In the cultivation system used here plenty of Fe is present in the medium. Next, evidently, in –Pi conditions, LPR1 alone has sufficient Fe²⁺ substrate available to do its job, even without CRR (or without HYP1) and possibly without both of them (?).

Considering that plants grown in the agar-based media are essentially supplied with oxidized Fe³⁺ (in the form of Fe-EDTA), and that LPR1 oxidizes Fe²⁺ to Fe³⁺, then we do assume that there must be a (or several) reductase(s) in roots providing the Fe²⁺ to LPR1, unless it is provided by a non-enzymatic route. While this was in the back of our mind when we started characterizing CRR, we realized later that CRR cannot be the enzyme providing directly Fe²⁺ to LPR1, because of the short-root phenotype of *crr*, as correctly stated by the reviewer. Instead, we envision that CRR works to modulate Fe³⁺/Fe²⁺ ratios (e.g. reducing Fe³⁺-malate) and impacts apoplastic Fe²⁺ levels by combining reduction with intracellular transport. We have thus modified the text of the manuscript to indicate this (page 5, first paragraph; page 22; page 24, last paragraph)

How would Fe³⁺-malate enhance ROS cycling? Essentially, we do not know and none of the papers published on the subject give a clear indication of this. Clearly, the presence of Fe³⁺-malate is tightly associated with ROS and the short-root phenotype, but how Fe³⁺-malate lead to enhanced ROS generation is unknown. Clearly this is an important subject for later studies. We would not be surprised if LPR1 ferredoxin activity would be closely associated with other enzymes to generate ROS, but this is very speculative at this point and beyond the scope of our study.

The need for LPR1 to generate Fe³⁺ to obtain the short-root phenotype under -Pi is indeed perplexing considering that there is abundant Fe³⁺ present from the media. In our view, it would favor a hypothesis where generation of Fe³⁺-malate needs to be generated *de novo*, likely in a specific location (apoplastic domain?) or/a enzyme complex, to generate the required ROS (implicating that external Fe³⁺ provided in the media cannot appropriately access this specific location or enzyme complex). If CRR has access to this specific location and/or enzymatic complex, then CRR expression would decrease the level of Fe³⁺ and ROS derived from it. Again, this is quite speculative at this point and beyond the scope of the current work, but we believe they are useful ideas to move forward. We have tried in the discussion (page 22) to hit the

right balance between providing hypothesis on how CRR and LPR1 may work while avoiding being too speculative.

9. Statement lines 492 to 494. The observations described/summarized here seem to contradict the present model of how primary root growth stalls under Pi deficiency. As the authors correctly state here, staining intensity was near saturation in Fig. S16 in both +Pi and +Pi 350 μ M Fe. Under -Pi, staining was even stronger (see Fig. 1I). The reviewer thus feels it is critical to provide data as shown in Fig. 6A and for the same genotypes, but grown in -Pi conditions.

The *crr* and *crr hyp1* mutants accumulate more iron than Col-0 at the root apical meristem under iron toxicity and phosphate deficiency. However, these are two different stresses and the consequences of different iron accumulation patterns on plant physiology can be completely different. Indeed, our data point towards this direction.

For the PersI-DAB staining shown in Figure 1i, although staining was strong, it was not clearly saturated since differences are present between *crr*, Col-0 and *pdr2* genotypes. However, we agree with the reviewer that the staining without DAB intensification could be useful. We thus performed the Perls staining in -Pi but unfortunately only the *pdr2* produced a noticeable signal under -Pi (Figure S12). Thus, in this case, in -Pi media without an excess source of Fe³⁺, intensification of Fe staining with DAB is needed to reveal a difference between Col-0 and *crr*.

10. Line 503 to 505: See my comment on replication, quantification and statistics. It would be very important to have data for iron uptake of roots and shoots also for *crr* and *crr hyp1* in addition to the wild type and the overexpressor. The differences observed in 6D could be a consequence merely of ectopic overexpression and thus have limited potential for informing us on the physiological role of *crr*. The statement in lines 511-513 on Fe accumulation in roots and shoots of *crr* and *crr hyp1* mutants is not sufficiently supported by data at present.

We now include data for iron uptake of roots and shoots for *crr* and *crr hyp1* in addition to the wild type and the overexpressor (Figure 6D). Given that there are no differences in the mutants, most likely due to the narrow expression pattern of these genes in the roots, we now soften the claim that *crr hyp1* has significantly less iron in shoots than the wt (page 21, second paragraph).

Comments on both manuscripts jointly:

11. The reviewer wonders whether by supplying ferricyanide (Fig. 4A and B) as a substrate to CRR proteins, a previously reported general action of ferricyanide in protein-bound heme oxidation comes to bear, and whether the observed ferrireductase activities in this manuscript are thus an observation that does not

correspond to the biological function of CRR. The reviewer wonders whether while CRR can donate an electron to Fe³⁺ under appropriately designed/altered conditions, this is in fact a negligible process in planta under the usual range of physiological conditions including phosphate deficiency. The authors of both manuscripts should contemplate (and possibly provide reasons against) the following model: HYP1 or CRR might donate an electron to directly quench extracellular ROS and thus Fenton chemistry in the apoplast of the stem cell niche. There needs to be a fine balance though, because in principle and under “permissive” conditions, electron donation to apoplastic molecules via CRR or HYP1 (e.g. in overexpressing shoots) can alternatively enhance ROS production and thus promote Fenton chemistry. Next, evidently, in –Pi conditions, LPR1/2 alone has sufficient Fe²⁺ substrate available to do its job, even without CRR (or without HYP1), and possibly without both of them (?), and thus LPR1/2 alone is able to account for the Fe-dependence of primary root growth inhibition phenotypes; the need for something additional (CRR/HYP1) to generate Fe²⁺ (in this case to moderate root growth inhibition) is not evident. Quantitative depletion of Fe(III) in the root apoplast through reduction to Fe²⁺ seems like a goal that is hard to achieve in a location that is permeable to the soil solution containing plenty of Fe(III).

The link between Fe and the primary root growth phenotypes associated with numerous mutants in this pathway (*lpr1*, *pdr1*, *almt1/stop1*, *crr*, *hyp1*) is clearly described in numerous publications and the current submitted manuscripts. Thus, the case for implicating Fe as the electron donor/acceptor in this pathway as well as ROS generation is quite strong. Having said this, as the reviewer indicates, the topic of ROS generation is very complex, with related pathways able to either generate or sequester ROS depending on the conditions. Although the case for the implication of CRR/HYP and LPR1 in Fe oxido-reduction is strong, we cannot exclude that other molecules may be involved, either as direct substrates for CRR/HYP1, or in secondary reactions following Fe reduction. These aspects are now discussed in more detail in the discussion (page 23, last paragraph).

As mentioned in point 8 above, we don't think that CRR function is to provide Fe²⁺ to LPR1. Instead CRR seems to reduce the amount of Fe³⁺ that could be chelated by malate. The reviewer reasoned that “Quantitative depletion of Fe(III) in the root apoplast through reduction to Fe²⁺ seems like a goal that is hard to achieve in a location that is permeable to the soil solution containing plenty of Fe(III)”. This sounds logical but so far, the experimental evidence points towards other direction since LPR1 is needed to regulate the primary root growth inhibition promoting Fe³⁺ production. In *lpr1* mutant, even though there is plenty of Fe³⁺ in the media, the root growth is insensitive to Pi deficiency, indicating that Fe³⁺ must be generated *de novo* by LPR1 to have its effect. We also find this observation rather counterintuitive and discuss in our reply to comment 8 above several hypothesis and research avenues that could be useful to explain these observations. Again, we have tried in the Discussion to hit the right balance between providing hypothesis on how CRR may work while avoiding being too speculative.

12. How do the authors explain the root growth difference in *crr1 almt1* (barely longer/meristem size not larger than *crr*, only about 50% root length of *almt1*) compared to *hyp1 almt1* (like *almt1*) in the accompanying manuscript under -Pi?

To try to clarify the matter, the Poirier and Giehl labs have exchanged seeds and grown them according to the protocol used in each respective lab. In our hands, *crr almt1* root length is shorter than *almt1*, but longer than *crr*. This intermediate phenotype can be explained by two factors: (1) *crr* introgression affected meristem maintenance in *almt1* mutant background, explaining why the double is shorter than *almt1*, and (2) CRR is not involved in cell elongation, explaining why the double mutant is longer than *crr*. In our hands, on -Pi, the *hyp1 almt1* double mutant has also shorter roots than *almt1* but longer than *hyp1*.

Data from Clúa and Poirier

In the Giehl's lab, the *almt1* mutant is fully epistatic to *hyp1*, the roots of the *hyp1 almt1* mutant being as long as the *almt1* mutant. However, the *crr* mutant is fully epistatic to *almt1*, the double mutant *crr almt1* having roots as short as the single *crr* mutant. Clearly, these results indicate that differences in experimental conditions used between the two labs are having an effect on the epistatic relationships with *almt1*. The protocols between the two labs differ in many aspects, including media composition (e.g. 1/6 MS versus 1/2 MS) and growth condition (e.g. plants grown directly on +/-Pi plates versus plants grown first on +Pi and then transferred to +/- Pi plates, light conditions). Obviously, the interactions between the mutant

alleles of *almt1*, *crr* and *hyp1* are complex, and likely involve several factors, including the influence of growth conditions as well as the expression pattern of the genes. We would note that Mora-Macias et al. (doi.org/10.1073/pnas.1701952114) and Balzergue et al., (DOI: 10.1038/ncomms15300) also showed discrepancies in the observed phenotypes for *almt1* under -Pi, with Balzergue and collaborators showing that *almt1* mutant accumulates Fe in the SCN to levels similar to Col-0, compromising the RAM maintenance, while Mora-Macias reported that this response was completely abolished in the *almt1* mutant. Such differences are also likely the consequence of different experimental conditions between their labs. It would thus appear that the phenotype of the *almt1* single mutant and double mutants derived from it would be very sensitive to the experimental set up. In contrast to this, the phenotype of the *crr* and *hyp1* single mutants, as well as the *crr hyp1* double, under + and -Pi conditions, is robust and consistent between the two labs, despite the differences in growth protocols.

We feel that fully dissecting the complexity associated with *ALMT1* is beyond the scope of our paper and that explaining all of these results in the paper would unnecessarily divert the attention of the reader away from the main aim of the paper, being the impact of CRR on root growth and Fe homeostasis. Thus we provide this information to the reviewer but have not included it in the paper.

Data from Maniero and Giehl

13. The authors should provide results from the root growth assay conducted under phosphate-sufficient and -deficient conditions also for the *crr hyp1* double mutant, in addition to the single mutants and the wild type. Including this in both manuscripts will contribute to the comparability of both manuscripts because the media and cultivation systems were different.

As suggested by the reviewer we now include in Figure S8 pictures and quantitative data of Col-0, *crr*, *hyp1*, and *crr hyp1* double mutant phenotypes under +Pi and -Pi. What we show is that while no difference is detectable between these mutants under +Pi condition, under -Pi there is a gradient in the length of primary root growth Col>*crr*>*hyp1*>*crr hyp1*>*pdr2*. In the companion manuscript of Maniero et al., although the media and growth conditions are distinct, the similar results were obtained, showing the robustness of the phenotypes.

14. The reviewer is struggling with the impression that plenty of Fe (see Fig. 2G, likely mostly Fe(III)) seems to be present near QC/SCN in +P conditions, but no stalling of primary root elongation occurs. While the presence of Fe is clearly necessary for this under -Pi conditions, the reviewer wonders whether much of what is observed several days later is merely a secondary and broader consequence of highly localized events that occurred within a much shorter period of time after the initiation of Pi deficiency.

We agree with the reviewer that it is somewhat puzzling that under +Pi condition, although Pearls-DAB staining does reveal adequate level of apoplastic Fe, primary root growth is not stalling. We cannot formerly exclude the hypothesis suggested above by the reviewer. It is also plausible that reduction in primary root growth observed under -Pi condition may not only be associated with more apoplastic Fe in absolute terms, but that the -Pi condition itself contributes to other changes that results in an increased sensitivity to apoplastic Fe level. The chemistry of the apoplast under -Pi may impact in what form/complex the Fe is present and how the plant cells reacts to it. For example, Muller et al, 2015 showed that under +Pi condition, apoplastic Fe is colocalized with P (implying that they form a complex), while no such co-localization is observed under -Pi condition. Thus Fe quantity is likely just one factor out of several that contributes to Fe impact on primary root growth. All of this is quite speculative at this point, but again useful to keep in mind while research on the topic is progressing.

15. The authors should consider in their models that the CYBDOM proteins may act somewhat independently in different parts of the plant – so in the opinion of the reviewer, if they generate an extracellular Fe²⁺ pool that is then taken up to contribute to shoot Fe levels, this Fe may not originate from the Fe²⁺ pool localized where stalling of root growth occurs under -Pi conditions. HYP1 appears to be present in a variety of cell types and even under +Pi conditions in a subset of these. This complicates pinning down the location of where the protein acts to contribute to foster primary root elongation under -Pi.

We agree with the reviewer that different CYBDOM members may act independently in different cells/tissues, and that the contribution of CYBDOMs to iron (and perhaps other metals like Cu) transport to shoots (or other tissues) might be independent of the localized effect that promotes root arrest under Pi deficiency. We included some wordings in this direction in the text (page 26). Clearly, detailed analysis of the pattern of expression of the various CYBDOMs as well as determining in which membranes they are expressed will be an important element for getting a clearer view of the impact and contribution of all these CYBDOM members to Fe homeostasis.

Minor comments:

16. Fig. S3A-B: transient localization of chimeric GFP-fusion proteins in a heterologous system is generally not as well accepted. The state-of-the art is to use the native promoter in stable transformant lines.

We agree with the statement of the reviewer. However, we have been facing particularly unusual circumstances regarding the expression of *CRR* using its endogenous promoter in a transgenic setting. As explained in page 11 (second paragraph), we did the experiment suggested by the reviewer but we could not detect any GFP expression, most likely because *CRR* possesses complex regulatory sequences that were not included in the *CRR* promoter region we cloned and/or higher-order chromatin organization or epigenetic signatures may play an important role in *CRR* promoter activity, and those may not be reproduced in a transgenic setting. We have made nearly 10 different constructs that included the *CRR* promoter, varying the length of the promoter from 200 bp to 5 kb, as well as using various terminators and binary vector backbones, without success in detecting either GFP or GUS when the *CRR* promoter was used. However, expression of *CRR-GFP* under the control of the CaMV35S promoter lead to complementation of the phenotypes of the Arabidopsis *crr* mutant. *CRR* is essentially localized at the plasma membrane in these complemented lines, and the same PM localization is observed in transiently transformed tobacco leaves (previously Figure S3, now S5).

Further evidence of the PM localization of *CRR* is provided by the fact that the line LPR1::*CRR-GFP* complementing the *crr* mutant phenotype also shows PM localization (Figure S14C; page 12, first paragraph).

17. Methods line 675: Please specify the type of agar used. The authors are asked to provide the information of how much phosphate and iron (μM in the final medium) the agar and the sugar components added to their media.

For all the experiments in agar we used Plant Agar from Duchefa. All the components of the media provided a final concentration of 20 μM of phosphate and 15 μM of iron. This information is now provided in material and methods (page 27).

18. The model (Fig. 6E) should clarify position(s) in the root. In the stem cell niche or in the root apical meristem, for example, the stele is not yet functional in root-to-shoot translocation yet. Could the authors also incorporate the action of CLE14/CLV2-PPR2 in the model or state a reason why this is omitted?

We have modified the figure to clarify these issues. Following the reviewer's suggestion we now included the CLE14 pathway in the model presented in Fig 6E. The text in the Discussion (page 24) have been modified accordingly.

19. Fig. S10: reviewer is surprised that there are no effects of pH and H₂O₂ levels on -Pi response of the wild type (Col-0). How is this compatible with the model?

There are differences in root length when Col-0 is grown under Pi deficiency under different pH and H₂O₂ levels. Those differences cannot be seen in former Figure S10 since, as indicated in the Y axis, the root length was normalized against Col-0 for each condition. We took that decision to make easier the comparison between genotypes rather than between conditions. Based on the reviewer's comment, we now think that other readers might have the same question. We now made a new version of Figure S18 showing the absolute values of root length, where it is evident that low pH and high concentrations of H₂O₂ further inhibits root development under Pi deficiency.

20. Lines 393-394: "show that CRR is" ☹ "show that CRR can act as"

Sentence has been modified as suggested (page 16, end of first paragraph).

21. Lines 423-424/and associated aspects: hard to grasp what interaction should mean; phrase by appropriate combinations of criteria for significant differences between genotypes AND between treatments?

Text has been modified to clarify this (page 17, second paragraph). Essentially, the analysis of "interaction" highlights how each genotype differs from Col-0 in their response to -Pi treatments.

22. Line 434: Please explain briefly the function of FRO8 here. Line 436: FRO4 not Fe-related (according to current knowledge)

We included a sentence describing the possible function of FRO8 (page 18, first paragraph). We also mention that FRO4 has been related to copper reduction and uptake, even though the specificity for Cu was not addressed (page 17, third paragraph). Similarly for FRO2 (and actually HYP1), most likely FRO4 activity is not limited to only one metal.

23. Line 396 says that transcriptomics were done on roots, but Figure legend 4D says transcriptomics were done on plants. Please clarify.

Transcriptomics were done on roots; we now specify this in the legend of figure 4D.

24. FeCN: use or define at least once correct chemical formula. $[\text{Fe}(\text{III})(\text{CN})_6]^{3-}$?

As suggested FeCN was defined as $[\text{Fe}(\text{III})(\text{CN})_6]^{3-}$ (page 15, first paragraph).

25. $\text{Fe}^{2+} \rightarrow \text{Fe}^{2+}$

We systematically made this mistake for Fe^{2+} and Fe^{3+} . We corrected this throughout the text.

26. Line 31: "CRR" should be in italics

CRR is now in italics in the abstract.

27. Line 86: not FTI but FIT instead

Typo was corrected (page 4, first paragraph).

28. Lines 121/122: please explain briefly in which direction calnexins influence Fe-dependent inhibition of primary root growth under Pi deficiency

A sentence was added to clarify this in the last paragraph of the Introduction (page 5, last paragraph)

29. Methods: add references for Perls/DAB

This has been done (page 31, second paragraph).

30. Line 235: 20% smaller meristem size (where is this shown?) Fig1DE?

It is shown in figure 1E. We included the specification (page 10, first paragraph).

31. Line 275: Abbreviation "SNC" must be defined; reviewer believes that this is probably a typo (should be "SCN")

Typo was fixed and SCN defined as stem cell niche (QC+initials) (page 11, first paragraph).

32. Line 549: adjust parentheses

Parenthesis are added automatically by the reference software. This problem will be solved at the editing stage.

33. Some of the information on numbers of replicates (" \geq ") is not compatible with this journal's instructions, to my knowledge.

We have modified the bar graphs so that the number of replicates appears as dots in each bar.

34. Indicate what was normalized to in the Legend of Fig. S7A.

The data was normalized to the Col-0 +Pi condition. As suggested we now specify in the legend (now Figure S13).

35. Mark stem cell niche cluster more explicitly in Fig. S7C

Done now in Supplemental Figure S13c. The stem cell niche includes the QC and initials.

36. Figure S8A: first and third rows, third column “endodermis”.

Corrected, now Figure S14

37. Fig. S11 title: a “to” is missing.

Corrected, now Figure S19.

38. Fig. S14D: add statistics?

No significant differences were found between genotypes. Now it is explicit in the legend, now Figure S25D.

39. Fig. S15D: y axis title “weight”

Corrected, now Figure S28.

Reviewer #4 (Remarks to the Author):

growth under Pi deficiency. They generated a *cnx1/cnx2* double mutant, and they did a proteomic analysis to compare the proteomic changes between Col-0 and the mutant under Pi deficiency. After the identification of CRR, they further generated a *crr* mutant and the *crr* mutant displayed an enhanced reduction of primary root growth, which is linked with increased accumulation of Fe in the root meristem. Additionally, overexpression of CRR showed a minus Pi insensitive phenotype and regulated the genes in Fe homeostasis. I have some comments regarding this manuscript.

1. all the MS data and MaxQuant search results are needed to deposit into a public repository.

As suggested, proteomic data was deposited at ProteomeXchange Consortium (identifier PXD040981). This information is provided in the section “Accession numbers” of the Material and Methods, with the statement “The proteomic data from this work has been deposited at ProteomeXchange Consortium via the PRIDE partner repository (<http://www.ebi.ac.uk/pride>) with the dataset identifier PXD040981”.

This data will be publicly available at the time of publication. In the meantime, the reviewers can access the data with the following access codes:

Reviewer account details:

Login : <http://www.ebi.ac.uk/pride>

Username: reviewer_pxd040981@ebi.ac.uk

Password: j8FhrjV

2. According to the proteomic analysis, CRR protein level in Col-0 roots did not display a significant change in response to Pi deficiency. If CRR is an important enzyme that regulates root growth under Pi deficiency, why the protein level did not increase under Pi deficiency? In Figure S4B, the mRNA level of CRR increased in Col-0 – Pi condition compared to +Pi condition. How do the authors explain this inconsistent result of mRNA and protein level?

Our proteomic and transcriptomic data were done using only root tissue, and they consistently show that CRR protein or mRNA levels do not change in response to phosphate deficiency. The quantification of *CRR* levels by qRT-PCR showed in Figure S4B (now S6B) was done using whole seedlings (roots plus shoots), and in this case, CRR showed an induction under Pi deficiency most likely because it is induced in shoots. Consistently, when we performed pRT-PCRs in roots and shoots separately (former Figure S7, now Figure S13A), we could determine that *CRR* transcript levels are induced upon Pi deficiency in shoots but not in roots. To avoid ambiguities, we now specify in the legend of Figure S4B (now Figure S6B) that roots and shoots were used to extract RNA.

Furthermore, the level of CRR protein expression does not need to change under -Pi conditions in order to play a role in root growth under these conditions. For example, changes in apoplastic Fe availability may be a key factor, or CRR activity may be modulated by post-translational process (e.g. phosphorylation).

3. Inconsistent result in Col-0 plus and minus Pi between transcriptomic and proteomic data. FRO4 showed increased in -Pi at transcriptional level, but FRO4 displayed a reduced protein level in -Pi compared to +pi in WT. How do the authors justify this result?

FRO4 is significantly (but weakly) repressed under Pi deficiency in our proteomic data (Table S1 shows a Log2FC of 0.45 decrease in WT -Pi versus WT +Pi) and transcriptomic data in Figure 4H shows a Log2FC of 1.87 increase in Col0 -Pi compared to Col-0 + Pi. Thus, even though FRO4 mRNA steady-state levels are higher under Pi deficiency, protein levels showed a reduction. This interesting observation most likely reflects a post-transcriptional regulation of FRO4 which has not been observed or studied before. The heterodirectionality in the regulation of transcript and protein levels have been extensively studied and many genes show this behavior upon different stresses. (e.g. [10.1371/journal.pgen.1001393](https://doi.org/10.1371/journal.pgen.1001393), doi.org/10.1038/s41586-020-2094-2).

REVIEWERS' COMMENTS

Reviewer #1 (Remarks to the Author):

This manuscript has been substantially improved, and I have no further comment

Reviewer #2 (Remarks to the Author):

I am satisfied with the authors' responses to my questions and comments and with this new version of the manuscript.

I have only two minor comments to make.

Line 383:

The sentence « PDR2 is epistatic to CRR » should be changed to something like « the *pdr2-1* mutation is epistatic to CRR OE » or « the root phenotype induced by the *pdr2-1* mutation masks that conferred by CRR overexpression ».

Lines 390-404 and Sup Fig. 18a:

The authors should keep in mind that increasing the pH of the growth medium represses the abundance of STOP1 and the expression of *ALMT1* (Balzergue et al. 2017; Godon et al. 2019). This lower expression of *ALMT1* probably also contributes to better root growth of Col-0 and *crr* under $-Pi$, reported in Sup Fig.18a.

Reviewer #3 (Remarks to the Author):

Most of my previous comments have been adequately addressed, and the text is also improved. I am sorry that the authors misunderstood some of my previous comments. I was meaning also for these that the essence or the most important part of the authors' responses should be included in the manuscript text whereas in some of their answers, they only replied to my comments (but made no changes in the manuscript).

1. My previous comment 1 (reviewer 3): The reviewer feels that the authors should highlight which of the results might be influenced by ectopic expression of CRR in both roots and shoots. Readers should be alerted to this in the manuscript text, at least when it first becomes biologically relevant (maybe Fig 2d, if judged by comparison to Fig. 1g) and also when RNA-seq data are described (because the reviewer feels that it is of particular relevance there and for the ^{55}Fe data).

2. My previous comment 1, later: To my understanding, the legend of Fig. 4 does not indicate that there was a correction of p-values for multiple comparisons. If it was done, then the legend text should be clarified. If it wasn't done, it should be done – irrespective of the consequences for the statistical significance of the differences.

3. Line 905-7 (following up on my previous comment 5): “After this, the root and shoot tissues were harvested separately and washed with extensively with 1/6-strength MS supplemented with 600 μM Fe(III)-EDTA.” Please delete the “with” after “washed”. Please add here: How exactly (moving, not moving, in how much volume, and – most importantly – for how much time? Reviewer believes that data are only fully reliable if the time period for this wash was sufficiently long and better also standardized across all samples. As this is not

going to be re-done (and the experiment was not overly insightful either), all the authors should do here is add this information. In addition, the authors should mention in the results text or in the methods that at least in roots, any ^{55}Fe precipitated onto cell walls outside cells is likely not to have been remobilized during the wash in the way it was carried out. The reviewer feels that this information will be very important for readers to fully understand the results.

4. Following up on my comment no. 6: It appears that the results of the statistics for the ^{55}Fe uptake (three diagrams in Fig. 6) were not corrected for multiple comparisons. If this is true, please correct and adjust characters as needed. Else please adjust legend text.

5. The reviewer understands that the numbers in Fig. 4 h,i,j are Log₂ FC, but could the authors please mention this explicitly in the figure legend? Where the comparison is specified via “x / y”, this is fairly clear to the reviewer, but where the authors write “Interaction (...)”, they should also specify in the legend how they calculated the numbers shown for the interactions and what these numbers are or reflect.

6. Line 511: Authors should add a brief remark in the results or the discussion to inform the readers that while IRT1, FRO2, bHLH100 and bHLH101 transcript levels are upregulated in WT plants under Fe deficiency usually, VTL2 transcript levels are downregulated under Fe deficiency. In this work, transcript levels of all these genes, when de-regulated, are de-regulated in the same direction, which is a bit peculiar.

7. Line 681: “The current work shows that CRR is a CYBDOM protein possessing ascorbate-mediated Fe³⁺ reductase activity.” Reviewer feels this needs to be toned down because no direct evidence for this was obtained here employing, for example, the purified protein: e.g. “is consistent with” instead of “shows”. This would also be more compatible with the text that follows.

8. In Fig. 6, where the authors show the scheme explaining the ^{55}Fe uptake experiment, they should use proper superscript and standardized designation for this radioisotope, and they should write “ μM ” instead of “uM”

9. Line 401 „enhanced the reduction“ – think about phrasing differently? Statement may be perceived as contradictory.

10. Line 491: “copper”, not “cooper”

Reviewer #4 (Remarks to the Author):

My questions have been appropriately addressed by the authors.

We appreciate the comments and suggestions from all reviewers. Below, you will find a point-by-point response to all the comments in red font. We have also highlighted the changes made in the manuscript text in a PDF file with the suffix “highlighted”. The line numbers correspond to this later PDF file.

Reviewer Comments:

Reviewer #1 (Remarks to the Author):

This manuscript has been substantially improved, and I have no further comment

Reviewer #2 (Remarks to the Author):

I am satisfied with the authors' responses to my questions and comments and with this new version of the manuscript.

I have only two minor comments to make.

Line 383:

The sentence « PDR2 is epistatic to CRR » should be changed to something like « the *pdr2-1* mutation is epistatic to CRR OE » or « the root phenotype induced by the *pdr2-1* mutation masks that conferred by CRR overexpression ».

The sentence was changed as suggested to « the *pdr2-1* mutation is epistatic to CRR OE », line 380

Lines 390-404 and Sup Fig. 18a:

The authors should keep in mind that increasing the pH of the growth medium represses the abundance of STOP1 and the expression of ALMT1 (Balzergue et al. 2017; Godon et al. 2019). This lower expression of ALMT1 probably also contributes to better root growth of Col-0 and *crr* under $-P_i$, reported in Sup Fig.18a.

We have added a line at the end of this paragraph to mention the potential contribution of pH on *ALMT* expression (lines 401-402).

Reviewer #3 (Remarks to the Author):

Most of my previous comments have been adequately addressed, and the text is also improved. I am sorry that the authors misunderstood some of my previous comments. I was meaning also for these that the essence or the most important part of the authors' responses should be included in the manuscript text

whereas in some of their answers, they only replied to my comments (but made no changes in the manuscript).

1. My previous comment 1 (reviewer 3): The reviewer feels that the authors should highlight which of the results might be influenced by ectopic expression of *CRR* in both roots and shoots. Readers should be alerted to this in the manuscript text, at least when it first becomes biologically relevant (maybe Fig 2d, if judged by comparison to Fig. 1g) and also when RNA-seq data are described (because the reviewer feels that it is of particular relevance there and for the ⁵⁵Fe data).

We have added some text when first describing the results from lines overexpressing *CRR* on the root phenotype (Figure 2), as well as when describing the RNA seq results (Figure 4d), to alert the reader that the results may be coming from either the higher expression level of *CRR* in cells normally expressing the endogenous *CRR* or its ectopic expression (lines 334-337, and 455-456).

2. My previous comment 1, later: To my understanding, the legend of Fig. 4 does not indicate that there was a correction of p-values for multiple comparisons. If it was done, then the legend text should be clarified. If it wasn't done, it should be done – irrespective of the consequences for the statistical significance of the differences.

For all the multiple comparisons made in our work, p-values were corrected for multiple comparisons using statistical hypothesis testing when performing one- or two-way ANOVA. This is now made explicitly in the “Statistics and reproducibility” section, lines 984-985.

3. Line 905-7 (following up on my previous comment 5): “After this, the root and shoot tissues were harvested separately and washed with extensively with 1/6-strength MS supplemented with 600 μM Fe(III)-EDTA.” Please delete the “with” after “washed”.

Typo was corrected

Please add here: How exactly (moving, not moving, in how much volume, and – most importantly – for how much time? Reviewer believes that data are only fully reliable if the time period for this wash was sufficiently long and better also standardized across all samples. As this is not going to be re-done (and the experiment was not overly insightful either), all the authors should do here is add this information. In addition, the authors should mention in the results text or in the methods that at least in roots, any ⁵⁵Fe precipitated onto cell walls outside cells is likely not to have been remobilized during the wash in the way it was carried out. The reviewer feels that this information will be very important for readers to fully understand the results.

We added these details in material and methods section, lines 907-910.

4. Following up on my comment no. 6: It appears that the results of the statistics for the ^{55}Fe uptake (three diagrams in Fig. 6) were not corrected for multiple comparisons. If this is true, please correct and adjust characters as needed. Else please adjust legend text.

For all the multiple comparisons made in our work, p-values were corrected for multiple comparisons using statistical hypothesis testing when performing one- or two-way ANOVA. This is now made explicitly in the “Statistics and reproducibility” section, lines 984-985.

5. The reviewer understands that the numbers in Fig. 4 h,i,j are Log₂ FC, but could the authors please mention this explicitly in the figure legend? Where the comparison is specified via “x / y”, this is fairly clear to the reviewer, but where the authors write “Interaction (...)”, they should also specify in the legend how they calculated the numbers shown for the interactions and what these numbers are or reflect.

On line 1448 of the legend to Figure 4, we have explicitly mentioned that the “numerical values in h-j represent log₂ fold-change”. We have also clearly defined in the same legend the term Interaction, namely “for *crr-1* as (*crr-1* -Pi / Col-0 -Pi) / (*crr-1* +Pi / Col-0 +Pi) and for CRR OE as (CRR OE -Pi / Col-0 -Pi) / (CRR OE +Pi / Col-0 +Pi), respectively.”

6. Line 511: Authors should add a brief remark in the results or the discussion to inform the readers that while *IRT1*, *FRO2*, *bHLH100* and *bHLH101* transcript levels are upregulated in WT plants under Fe deficiency usually, *VTL2* transcript levels are downregulated under Fe deficiency. In this work, transcript levels of all these genes, when de-regulated, are de-regulated in the same direction, which is a bit peculiar.

Lines 510-513, we have modified the text to make these changes in gene expression explicit: “In addition, a number of Fe-related genes were mildly deregulated in CRR OE. Thus, the *bHLH101*, *FRO2* and *IRT1* genes that are normally activated by Fe deficiency, as well as the *VTL2* typically repressed under similar condition, were all slightly upregulated in CRR OE.”

7. Line 681: “The current work shows that CRR is a CYBDOM protein possessing ascorbate-mediated Fe³⁺ reductase activity.” Reviewer feels this needs to be toned down because no direct evidence for this was obtained here employing, for example, the purified protein: e.g. “is consistent with” instead of “shows”. This would also be more compatible with the text that follows.

We changed the sentence (line 681-682) to: “The current work shows that CRR is a CYBDOM protein with features consistent with an ascorbate-dependent Fe³⁺ reductase.”

8. In Fig. 6, where the authors show the scheme explaining the ^{55}Fe uptake experiment, they should use proper superscript and standardized designation for this radioisotope, and they should write “ μM ” instead of “uM”

We corrected the typos in Figure 6.

9. Line 401 „enhanced the reduction“ – think about phrasing differently? Statement may be perceived as contradictory.

We changed the sentence to “it exacerbated *crr-1* root growth inhibition under –Pi”, line 398

10. Line 491: “copper”, not “cooper”

Typo was corrected, line 490.

Reviewer #4 (Remarks to the Author):

My questions have been appropriately addressed by the authors.